# Neural Multi-Objective Combinatorial Optimization with Diversity Enhancement

**Jinbiao Chen**[1], **Zizhen Zhang**[1], **Zhiguang Cao**[2], **Yaoxin Wu**[3], **Yining Ma**[4],
**Te Ye**[1], and **Jiahai Wang**[1,5,6,*]

[1]School of Computer Science and Engineering, Sun Yat-sen University, P.R. China
[2]School of Computing and Information Systems, Singapore Management University, Singapore
[3]Department of Industrial Engineering & Innovation Sciences,
Eindhoven University of Technology, Netherlands
[4]Department of Industrial Systems Engineering & Management,
National University of Singapore, Singapore
[5]Key Laboratory of Machine Intelligence and Advanced Computing, Ministry of Education,
Sun Yat-sen University, P.R. China
[6]Guangdong Key Laboratory of Big Data Analysis and Processing, Guangzhou, P.R. China
`chenjb69@mail2.sysu.edu.cn, zhangzzh7@mail.sysu.edu.cn`
`zgcao@smu.edu.sg, y.wu2@tue.nl, yiningma@u.nus.edu`
`yete@mail2.sysu.edu.cn, wangjiah@mail.sysu.edu.cn`

## Abstract

Most of existing neural methods for multi-objective combinatorial optimization (MOCO) problems solely rely on decomposition, which often leads to repetitive solutions for the respective subproblems, thus a limited Pareto set. Beyond decomposition, we propose a novel neural heuristic with diversity enhancement (NHDE) to produce more Pareto solutions from two perspectives. On the one hand, to hinder duplicated solutions for different subproblems, we propose an indicator-enhanced deep reinforcement learning method to guide the model, and design a heterogeneous graph attention mechanism to capture the relations between the instance graph and the Pareto front graph. On the other hand, to excavate more solutions in the neighborhood of each subproblem, we present a multiple Pareto optima strategy to sample and preserve desirable solutions. Experimental results on classic MOCO problems show that our NHDE is able to generate a Pareto front with higher diversity, thereby achieving superior overall performance. Moreover, our NHDE is generic and can be applied to different neural methods for MOCO.

## 1 Introduction

Multi-objective combinatorial optimization (MOCO) has been extensively studied in the communities of computer science and operations research [1, 2]. It also commonly exists in many industries, such as transportation [3], manufacturing [4], energy [5], and telecommunication [6]. MOCO features multiple conflicting objectives based on NP-hard combinatorial optimization (CO), practical yet more complex. Rather than finding an optimal solution like in the single-objective optimization, MOCO pursues a set of Pareto-optimal solutions, called *Pareto set*, to trade-off the multiple objectives. In general, a decent Pareto set is captured by both desirable convergence (optimality) and diversity.

Since exactly solving MOCO may require exponentially increasing computational time [7], the heuristic methods [8] have been favored in practice over the past few decades, which aim to yield an

---

[*]Corresponding Author.

37th Conference on Neural Information Processing Systems (NeurIPS 2023).

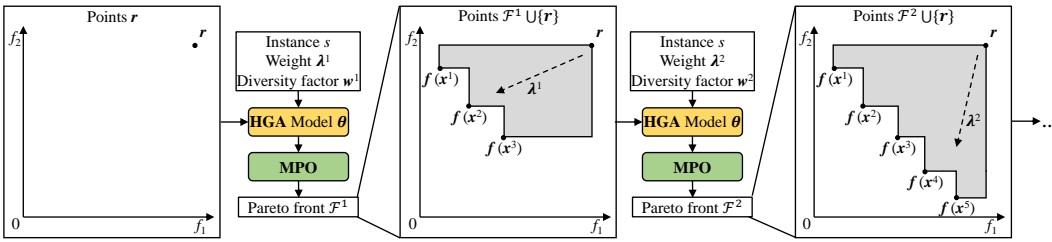

Figure 1: The framework of NHDE. For subproblem $i$, the heterogeneous graph attention (HGA) model takes instance $s$, *points* $\mathcal{F}^{i-1} \cup \{r\}$, weight $\lambda^i$, and diversity factor $w^i$ as inputs, and generates solutions to optimize the scalar objective and hypervolume (size of the gray area). More solutions are sampled and the Pareto front $\mathcal{F}^i$ is then efficiently updated based on multiple Pareto optima (MPO).

approximate Pareto set. Despite a relatively high efficiency, heuristic methods need domain-specific knowledge and massive iterative search. As *neural CO* methods based on deep reinforcement learning (DRL) recently achieved notable success in CO problems such as routing, scheduling, and bin packing [9–12], a number of *neural MOCO* methods have also been accordingly developed [13–16]. Typically, parameterized as a deep model, Neural MOCO is able to automatically learn a heuristic (or policy), so as to directly construct near-optimal solutions in an end-to-end fashion, which takes much less computational time than the traditional ones.

Existing neural MOCO methods mostly decompose an MOCO problem into a series of single-objective CO subproblems and derive a Pareto set by solving them. However, while enjoying a favorable efficiency for neural MOCO, the sole decomposition is less effective in finding as many diverse solutions as possible, since constructing an optimal solution for the decomposed subproblems is always carried out independently, causing repetitive or duplicated ones for different subproblems.

To tackle this issue, we propose a neural heuristic with diversity enhancement (NHDE), as illustrated in Figure 1. Distinguished from existing neural methods, NHDE couples the decomposition with a comprehensive indicator to learn a policy that can produce diverse solutions across the subproblems while further improving the performance. Besides, for a given subproblem, multiple relevant solutions, rather than a single optimal solution with respect to the scalar objective, are found based on a proposed multiple Pareto optima (MPO) strategy, so as to further strengthen the diversity.

Our contributions are summarized as follows. (1) We propose an indicator-enhanced DRL method. To encourage the deep model to generate different yet diverse solutions for decomposed subproblems, NHDE inputs the Pareto front composed of the preceding solutions and introduces an indicator comprehensively measuring convergence and diversity in the reward. (2) We design a heterogeneous graph attention model to effectively capture the correlation between an instance graph and its Pareto front graph. (3) We present a multiple Pareto optima (MPO) strategy to further identify more relevant solutions in the neighborhood of each subproblem and efficiently update the Pareto front. (4) We deploy our NHDE with two different types of neural MOCO methods to demonstrate its versatility. Experimental results based on various MOCO problems show that our NHDE outperforms state-of-the-art neural baselines, especially with significant improvement in the diversity.

## 2 Related works

**Exact and heuristic methods for MOCO.** Exact methods [7, 17] for MOCO can attain the accurate Pareto set, but their computational time may grow exponentially, rendering them less practical. As an alternative, heuristic methods such as multi-objective evolutionary algorithms (MOEAs) have gained widespread attention in practice. Dominance-based NSGA-II [18], decomposition-based MOEA/D [19], and indicator-based SMS-EMOA [20] are three typical paradigms of MOEAs. In these MOEAs, local search, a crucial technique specialized to the target CO, is usually employed [21–23].

**Neural CO.** In the past few years, neural *construction* methods [24–26] were proposed to rapidly yield high-quality solutions in an end-to-end fashion. A well-known representative is *Attention Model* (AM) [27], which was developed based on the *Transformer* architecture [28]. Then, AM inspired a large number of subsequent works [29–35], which further boosted the performance. Among them,

the policy optimization with multiple optima (POMO) [36], leveraging the solution and problem symmetries, is recognized as a prominent approach. Besides, the other line of works, known as neural *improvement* methods [37–41], exploited DRL to assist the iterative improvement process from an initial but complete solution, following a learn-to-improve paradigm.

**Neural MOCO.** Decomposition is a mainstream scheme in learning-based methods for multi-objective optimization [42–44]. An MOCO problem can be decomposed into a series of single-objective CO problems and then solved by neural construction methods to approximate the Pareto set. A couple of preliminary works trained multiple deep models with transfer learning [13, 45], where each deep model coped with one subproblem. Evolutionary learning [46, 47] was introduced to evolve deep models to further improve the performance. Instead of training multiple deep models for preset weights, preference-conditioned multi-objective combinatorial optimization (PMOCO) [14] and Meta-DRL (MDRL) [15], both of which only trained one deep model, were more flexible and practical. For a given weight vector, the former used a hypernetwork to derive the decoder parameters to solve the corresponding subproblem, while the latter rapidly fine-tuned a pre-trained meta-model to solve the corresponding subproblem. However, those solely decomposition-based neural MOCO methods are limited in the diversity with respect to the solutions of the Pareto set, since some subproblems may lead to duplicated solutions, especially when they are solved independently.

## 3  Preliminary

### 3.1  MOCO

An MOCO problem with $M$ objectives can be expressed as $\min_{\boldsymbol{x} \in \mathcal{X}} \boldsymbol{f}(\boldsymbol{x}) = (f_1(\boldsymbol{x}), f_2(\boldsymbol{x}), \ldots, f_M(\boldsymbol{x}))$, where $\mathcal{X}$ is a set of discrete decision variables.

**Definition 1 (Pareto dominance).** A solution $\boldsymbol{x}^1 \in \mathcal{X}$ dominates another solution $\boldsymbol{x}^2 \in \mathcal{X}$ ($\boldsymbol{x}^1 \prec \boldsymbol{x}^2$), if and only if $f_i(\boldsymbol{x}^1) \leq f_i(\boldsymbol{x}^2), \forall i \in \{1, \ldots, M\}$ and $\exists j \in \{1, \ldots, M\}, f_j(\boldsymbol{x}^1) < f_j(\boldsymbol{x}^2)$.

**Definition 2 (Pareto optimality).** A solution $\boldsymbol{x}^* \in \mathcal{X}$ is Pareto-optimal if it is not dominated by any other solution, i.e., $\nexists \boldsymbol{x}' \in \mathcal{X}$ such that $\boldsymbol{x}' \prec \boldsymbol{x}^*$.

**Definition 3 (Pareto set/front).** MOCO aims to uncover a Pareto set, comprising all Pareto optimal solutions $\mathcal{P} = \{\boldsymbol{x}^* \in \mathcal{X} \mid \nexists \boldsymbol{x}' \in \mathcal{X} : \boldsymbol{x}' \prec \boldsymbol{x}^*\}$. The Pareto front $\mathcal{F} = \{\boldsymbol{f}(\boldsymbol{x}) \mid \boldsymbol{x} \in \mathcal{P}\}$ corresponds to the objective values of Pareto set, with each $\boldsymbol{f}(\boldsymbol{x})$ referred to as a *point* in the objective space.

### 3.2  Decomposition

For MOCO, decomposition [19] is a prevailing scheme due to its flexibility and effectiveness. An MOCO problem can be decomposed into $N$ subproblems with $N$ weights. Each subproblem is a single-objective CO problem via scalarization $g(\boldsymbol{x}|\boldsymbol{\lambda})$ with a weight $\boldsymbol{\lambda} \in \mathcal{R}^M$ satisfying $\lambda_m \geq 0$ and $\sum_{m=1}^{M} \lambda_m = 1$. The Pareto set then can be derived by solving the $N$ subproblems.

The simplest yet effective decomposition approach is the weighted sum (WS). It uses the linear scalarization of $M$ objectives, which hardly raises the complexity of the subproblems, as follows,

$$\min_{\boldsymbol{x} \in \mathcal{X}} g_{\mathrm{ws}}(\boldsymbol{x}|\boldsymbol{\lambda}) = \sum_{m=1}^{M} \lambda_m f_m(\boldsymbol{x}). \tag{1}$$

### 3.3  Indicator

Hypervolume (HV) is a mainstream indicator to measure performance, as it can comprehensively assess the convergence and diversity without the exact Pareto front [48]. For a Pareto front $\mathcal{F}$ in the objective space, $\mathrm{HV}_{\boldsymbol{r}}(\mathcal{F})$ with respect to a fixed reference point $\boldsymbol{r} \in \mathcal{R}^M$ is defined as follows,

$$\mathrm{HV}_{\boldsymbol{r}}(\mathcal{F}) = \mu \left( \bigcup_{\boldsymbol{f}(\boldsymbol{x}) \in \mathcal{F}} [\boldsymbol{f}(\boldsymbol{x}), \boldsymbol{r}] \right), \tag{2}$$

where $\mu$ is the Lebesgue measure, i.e., $M$-dimensional volume, and $[\boldsymbol{f}(\boldsymbol{x}), \boldsymbol{r}]$ is a $M$-dimensional cube, i.e., $[\boldsymbol{f}(\boldsymbol{x}), \boldsymbol{r}] = [f_1(\boldsymbol{x}), r_1] \times \cdots \times [f_M(\boldsymbol{x}), r_M]$.

A 2-dimensional example with 5 *points* in the objective space is depicted in Figure 1, where $\mathcal{F} = \{\boldsymbol{f}(\boldsymbol{x}^1), \boldsymbol{f}(\boldsymbol{x}^2), \boldsymbol{f}(\boldsymbol{x}^3), \boldsymbol{f}(\boldsymbol{x}^4), \boldsymbol{f}(\boldsymbol{x}^5)\}$. $\mathrm{HV}_{\boldsymbol{r}}(\mathcal{F})$ is equal to the size of the gray area, and finally normalized into $[0, 1]$. All methods share the same reference point $\boldsymbol{r}$ for a problem (see Appendix A).

## 4 Methodology

Our *neural heuristic with diversity enhancement* (NHDE) exploits indicator-enhanced DRL to produce diverse solutions across different subproblems and leverages a multiple Pareto optima (MPO) strategy to find multiple neighbor solutions for each subproblem, as illustrated in Figure 1. Specifically, an MOCO problem is decomposed into $N$ single-objective subproblems with $N$ weights, which are solved dependently by a unified heterogeneous graph attention (HGA) model $\boldsymbol{\theta}$. For each subproblem $i$, its features together with the current Pareto front $\mathcal{F}$ (*points* in the objective space) constituted by preceding solutions are input to model $\boldsymbol{\theta}$, which is guided by the scalar objective with the HV indicator. Then, MPO is utilized to sample multiple solutions and efficiently update $\mathcal{F}$.

### 4.1 Indicator-enhanced DRL

Given a problem instance $s$, we sequentially solve its subproblem $i \in \{1, \ldots, N\}$, each associated with weight $\boldsymbol{\lambda}^i$. Let $\boldsymbol{\pi}^i = \{\pi_1^i, \ldots, \pi_T^i\}$ denote the obtained solution[2] at step $i$, and let $\mathcal{F}^i$ be the Pareto front yielded by solutions from subproblem 1 to $i$. In each step $i$, we select up to $K$ top *points* $\boldsymbol{f}(\boldsymbol{\pi}) \in \mathcal{F}^{i-1}$ from the Pareto front at step $i-1$ with the ranking determined by the scalar objective $g(\boldsymbol{\pi}|s, \boldsymbol{\lambda}^i)$ with respect to the new given weight $\boldsymbol{\lambda}^i$. The corresponding scalar objective and the induced surrogate landscape $\tilde{\mathcal{F}}^{i-1} \subseteq \mathcal{F}^{i-1}$ based on those selected solutions are treated as the policy network inputs (see Figure 2), so as to construct a new solution $\boldsymbol{\pi}^i$ and yield a new $\mathcal{F}^i$.

The construction of the solution $\boldsymbol{\pi}^i$ with length $T$ for each subproblem $i$ can be cast as a Markov decision process. In particular, 1) the *state* includes the weight $\boldsymbol{\lambda}^i$, user-defined diversity factor $\boldsymbol{w}^i \in \mathcal{R}^2$ satisfying $w_1^i, w_2^i \geq 0$ and $w_1^i + w_2^i = 1$, partial solution $\boldsymbol{\pi}_{1:t-1}^i$, instance $s$, and $\tilde{\mathcal{F}}_{\boldsymbol{r}}^{i-1}$, where $\tilde{\mathcal{F}}_{\boldsymbol{r}}^{i-1} = \tilde{\mathcal{F}}^{i-1} \cup \{\boldsymbol{r}\}$ incorporates the aforementioned surrogate landscape at step $i$ and the given reference *point* $\boldsymbol{r}$; 2) the *action* is to add a node $\pi_t^i$ into $\boldsymbol{\pi}_{1:t-1}^i$; 3) the *state transition* transforms $\boldsymbol{\pi}_{1:t-1}^i$ to $\boldsymbol{\pi}_{1:t}^i$, denoted as $\boldsymbol{\pi}_{1:t}^i = \{\boldsymbol{\pi}_{1:t-1}^i, \pi_t^i\}$; 4) the *reward* is defined as $R^i = -w_1^i \times g(\boldsymbol{\pi}^i|s, \boldsymbol{\lambda}^i) + w_2^i \times \mathrm{HV}_{\boldsymbol{r}}(\tilde{\mathcal{F}}^{i-1} \cup \{\boldsymbol{f}(\boldsymbol{\pi}^i)\})$, where we introduce the hypervolume $\mathrm{HV}_{\boldsymbol{r}}(\tilde{\mathcal{F}}^{i-1} \cup \{\boldsymbol{f}(\boldsymbol{\pi}^i)\})$ to guide the search; and 5) the stochastic *policy* generating the solution $\boldsymbol{\pi}^i$ is expressed as $P(\boldsymbol{\pi}^i|s, \tilde{\mathcal{F}}_{\boldsymbol{r}}^{i-1}, \boldsymbol{\lambda}^i, \boldsymbol{w}^i) = \prod_{t=1}^T P_{\boldsymbol{\theta}}(\pi_t^i|\boldsymbol{\pi}_{1:t-1}^i, s, \tilde{\mathcal{F}}_{\boldsymbol{r}}^{i-1}, \boldsymbol{\lambda}^i, \boldsymbol{w}^i)$, with the probability of node selection $P_{\boldsymbol{\theta}}(\pi_t^i|\boldsymbol{\pi}_{1:t-1}^i, s, \tilde{\mathcal{F}}_{\boldsymbol{r}}^{i-1}, \boldsymbol{\lambda}^i, \boldsymbol{w}^i)$ parameterized by a deep model $\boldsymbol{\theta}$.

We would like to note that our NHDE is generic, and can directly integrate the base model $\boldsymbol{\theta}$ with the existing decomposition-based neural MOCO methods. We demonstrate this property by applying it to two state-of-the-art methods, PMOCO [14] and MDRL [15], denoted as NHDE-P and NHDE-M, respectively. Given $\boldsymbol{\lambda}^i$ and $\boldsymbol{w}^i$ as inputs, NHDE-P uses a hypernetwork to generate the decoder parameters of the model $\boldsymbol{\theta}(\boldsymbol{\lambda}^i, \boldsymbol{w}^i)$, while NHDE-M fine-tunes the pre-trained meta model $\boldsymbol{\theta}_{\mathrm{meta}}$ with a few steps to address the corresponding subproblem. More details are presented in Appendix B.

### 4.2 Heterogeneous graph attention

To effectively solve subproblem $i$, the model should jointly capture the representations of both the instance's node graph and the Pareto front's *point* graph. Based on the encoder-decoder structure, we thus design a heterogeneous graph attention (HGA) model to correlate the two heterogeneous graphs as depicted in Figure 2. In the following section, we omit superscript $i$ for better readability.

**Encoder.** Given an instance graph $s$ containing $n$ nodes with $Z$-dimensional features and a Pareto front graph $\tilde{\mathcal{F}}_{\boldsymbol{r}}^{i-1}$ containing $k$ *points* ($k \leq K + 1$) with $M$-dimensional features, their initial embeddings $\boldsymbol{h}_1^{(0)}, \ldots, \boldsymbol{h}_n^{(0)} \in R^d$ and $\boldsymbol{g}_1^{(0)}, \ldots, \boldsymbol{g}_k^{(0)} \in R^d$ are yielded by a linear projection with trainable parameters $W_h$ and $W_g$, respectively, and $d$ is empirically set to 128. The eventual embeddings $\boldsymbol{h}_1^{(L)}, \ldots, \boldsymbol{h}_n^{(L)}$ and $\boldsymbol{g}_1^{(L)}, \ldots, \boldsymbol{g}_k^{(L)}$ are derived by further passing through $L = 6$ attention layers.

---

[2]In this sub-section, we consider the construction of only one solution in each step for better readability; however, we note that multiple solutions can be sampled and our formulation would work in a similar manner.

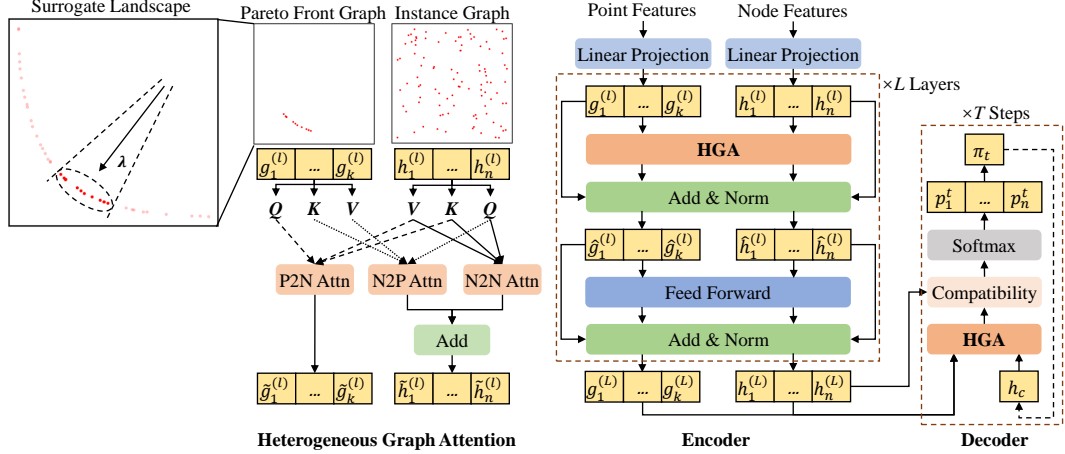

Figure 2: Illustration of the proposed heterogeneous graph attention (HGA) model.

Each attention layer is composed of a multi-head HGA layer with $Y = 8$ heads and a fully connected feed-forward sublayer. For layer $l \in \{1, \ldots, L\}$, HGA computes the representations $\tilde{\boldsymbol{h}}_u^{(l)}$ and $\tilde{\boldsymbol{g}}_u^{(l)}$ across the heterogeneous graphs, which are used to update the embeddings $\boldsymbol{h}_u^{(l)}$ and $\boldsymbol{g}_u^{(l)}$. Skip-connection [49] and batch normalization [50] are both adopted in each sublayer, as follows,

$$\boldsymbol{h}_u^{(l)} = \mathrm{BN}(\hat{\boldsymbol{h}}_u + \mathrm{FF}(\hat{\boldsymbol{h}}_u)), \ \hat{\boldsymbol{h}}_u = \mathrm{BN}(\boldsymbol{h}_u^{(l-1)} + \tilde{\boldsymbol{h}}_u^{(l)}), \quad \forall u \in \{1, \ldots, n\}, \tag{3}$$

$$\boldsymbol{g}_u^{(l)} = \mathrm{BN}(\hat{\boldsymbol{g}}_u + \mathrm{FF}(\hat{\boldsymbol{g}}_u)), \ \hat{\boldsymbol{g}}_u = \mathrm{BN}(\boldsymbol{g}_u^{(l-1)} + \tilde{\boldsymbol{g}}_u^{(l)}), \quad \forall u \in \{1, \ldots, k\}. \tag{4}$$

**Decoder.** The decoder, composed of a multi-head HGA layer and a *compatibility* layer, autoregressively constructs a solution according to the probability distribution with $T$ steps. At decoding step $t \in \{1, ..., T\}$, the *glimpse* $\boldsymbol{q}_c$ of the *context* embedding $\boldsymbol{h}_c$ (see Appendix C) is computed by the HGA layer. Then, the *compatibility* $\boldsymbol{\alpha}$ is calculated as follows,

$$\alpha_u = \begin{cases} -\infty, & \text{node } u \text{ is masked} \\ C \cdot \tanh(\frac{\boldsymbol{q}_c^T (W^K \boldsymbol{h}_u^{(L)})}{\sqrt{d/Y}}), & \text{otherwise} \end{cases} \tag{5}$$

where $C$ is set to 10 [27]. Finally, softmax is employed to calculate the selection probability distribution $P_{\boldsymbol{\theta}}(\boldsymbol{\pi}|s, \tilde{\mathcal{F}}_{\boldsymbol{r}}^{i-1}, \boldsymbol{\lambda}^i, \boldsymbol{w}^i)$ for nodes, i.e., $P_{\boldsymbol{\theta}}(\pi_t|\boldsymbol{\pi}_{1:t-1}, s, \tilde{\mathcal{F}}_{\boldsymbol{r}}^{i-1}, \boldsymbol{\lambda}^i, \boldsymbol{w}^i) = \mathrm{Softmax}(\boldsymbol{\alpha})$.

**HGA.** The HGA layer in the encoder captures three key relations between the node graph and the *point* graph. The first, *node-to-node* $\alpha_{uv}^{hh}$, indicates each node's attention towards others within the same instance to construct a promising solution. The second, *node-to-point* $\alpha_{uv}^{hg}$, suggests each node's attention to *points*, guiding the constructed solutions distinct from the existing ones in the current Pareto front. The third, *point-to-node* $\alpha_{uv}^{gh}$, indicates each *point*'s attention to nodes, facilitating the learning of the mapping from a solution to its objective values. We disregard the less meaningful *point-to-point* attention. Concretely, Eq. (6) defines the above three attention scores, which are separately normalized as $\tilde{\alpha}_{uv}^{hh}$, $\tilde{\alpha}_{uv}^{hg}$, and $\tilde{\alpha}_{uv}^{hg}$ by softmax. Then, $\tilde{\boldsymbol{h}}_u$ and $\tilde{\boldsymbol{g}}_u$ are computed by Eq. (7).

$$\alpha_{uv}^{hh} = \frac{(W_h^Q \boldsymbol{h}_u)^T (W_h^K \boldsymbol{h}_v)}{\sqrt{d/Y}}, \ \alpha_{uv}^{hg} = \frac{(W_h^Q \boldsymbol{h}_u)^T (W_g^K \boldsymbol{g}_v)}{\sqrt{d/Y}}, \ \alpha_{uv}^{gh} = \frac{(W_g^Q \boldsymbol{g}_u)^T (W_h^K \boldsymbol{h}_v)}{\sqrt{d/Y}}. \tag{6}$$

$$\tilde{\boldsymbol{h}}_u = \sum_{v=1}^n \tilde{\alpha}_{uv}^{hh} W_h^V \boldsymbol{h}_v + \sum_{v=1}^k \tilde{\alpha}_{uv}^{hg} W_g^V \boldsymbol{g}_v, \ \tilde{\boldsymbol{g}}_u = \sum_{v=1}^n \tilde{\alpha}_{uv}^{gh} W_h^V \boldsymbol{h}_v. \tag{7}$$

Finally, as for the multi-head attention, $\tilde{\boldsymbol{h}}_u$ and $\tilde{\boldsymbol{g}}_u$ are further computed as follows,

$$\tilde{\boldsymbol{h}}_u = W_h^O \mathrm{Concat}(\tilde{\boldsymbol{h}}_{u,1}, \ldots, \tilde{\boldsymbol{h}}_{u,Y}), \ \tilde{\boldsymbol{g}}_u = W_g^O \mathrm{Concat}(\tilde{\boldsymbol{g}}_{u,1}, \ldots, \tilde{\boldsymbol{g}}_{u,Y}), \tag{8}$$

where $\tilde{\boldsymbol{h}}_{u,y}$ and $\tilde{\boldsymbol{g}}_{u,y}$ for head $y \in \{1, ..., Y\}$ are obtained according to Eq. (7). In the multi-head HGA, $W_h^Q$, $W_h^K$, $W_h^V$, $W_h^O$, $W_g^Q$, $W_g^K$, $W_g^V$, and $W_g^O$ are independent trainable parameters. Similarly, in the decoder, the glimpse $\boldsymbol{q}_c$ is calculated by the context embedding $\boldsymbol{h}_c$ with the addition of the *context-to-node* and *context-to-point* attention, i.e., replacing $\boldsymbol{h}_u$ with $\boldsymbol{h}_c$ in the Eq. (6–8).

**Algorithm 1** Training algorithm of NHDE-P

---

1: **Input:** weight distribution $\Lambda$, diversity-factor distribution $\mathcal{W}$, instance distribution $\mathcal{S}$, number of training steps $E$, number of sampled weights per step $N'$, batch size $B$, instance size $n$
2: Initialize the model parameters $\boldsymbol{\theta}$
3: **for** $e = 1$ to $E$ **do**
4:    $s_i \sim$ **SampleInstance**($\mathcal{S}$)    $\forall i \in \{1, \cdots, B\}$
5:    Initialize $\mathcal{F}_i \leftarrow \emptyset$    $\forall i$
6:    **for** $n' = 1$ to $N'$ **do**
7:       $\boldsymbol{\lambda} \sim$ **SampleWeight**($\Lambda$)
8:       $\boldsymbol{w} \sim$ **SampleDiversityFactor**($\mathcal{W}$)
9:       $\boldsymbol{\pi}_i^j \sim$ **SampleSolution**($P_{\boldsymbol{\theta}(\boldsymbol{\lambda},\boldsymbol{w})}(\cdot|s_i, \tilde{\mathcal{F}}_{\boldsymbol{r},i}, \boldsymbol{\lambda}, \boldsymbol{w})$)    $\forall i \in \{1, \cdots, B\}$    $\forall j \in \{1, \cdots, n\}$
10:       $R_i^j \leftarrow -w_1 g(\boldsymbol{\pi}_i^j|s_i, \boldsymbol{\lambda}) + w_2 \text{HV}_{\boldsymbol{r}}(\tilde{\mathcal{F}}_i \cup \{\boldsymbol{f}(\boldsymbol{\pi}_i^j)\})$    $\forall i, j$
11:       $b_i \leftarrow \frac{1}{n} \sum_{j=1}^n (-R_i^j)$    $\forall i$
12:       $\nabla \mathcal{J}(\boldsymbol{\theta}) \leftarrow \frac{1}{Bn} \sum_{i=1}^B \sum_{j=1}^n [(-R_i^j - b_i)\nabla_{\boldsymbol{\theta}(\boldsymbol{\lambda},\boldsymbol{w})} \log P_{\boldsymbol{\theta}(\boldsymbol{\lambda},\boldsymbol{w})}(\boldsymbol{\pi}_i^j|s_i, \tilde{\mathcal{F}}_{\boldsymbol{r},i}, \boldsymbol{\lambda}, \boldsymbol{w})]$
13:       $\boldsymbol{\theta} \leftarrow$ **Adam**($\boldsymbol{\theta}, \nabla \mathcal{J}(\boldsymbol{\theta})$)
14:       $\mathcal{G}_i \leftarrow \{\boldsymbol{f}(\boldsymbol{\pi}_i^1), \ldots, \boldsymbol{f}(\boldsymbol{\pi}_i^n)\}$    $\forall i$
15:       $\mathcal{F}_i \leftarrow \text{MPO}(\tilde{\mathcal{F}}_i \cup \tilde{\mathcal{G}}_i)$    $\forall i$
16:    **end for**
17: **end for**
18: **Output:** The model parameter $\boldsymbol{\theta}$

---

### 4.3 Multiple Pareto optima strategy

Contrary to single-objective problems that focus on a single optimal solution, MOCO problems involve a series of Pareto-optimal solutions. In light of this, we introduce a multiple Pareto optima (MPO) strategy to uncover multiple solutions for each subproblem by leveraging the Pareto optimality.

When solving subproblem $i$, more than one solution can be attained by sampling, e.g., sampling with multiple start nodes as did in POMO [36]. In this case, $\mathcal{F}^i = \text{MPO}(\mathcal{F}^{i-1} \cup \mathcal{G}^i)$, where $\mathcal{G}^i$ contains all the candidate *points* (find by sampling) to be introduced in the new Pareto front and $\text{MPO}(\cdot)$ is an operator that updates the Pareto front. However, as the complexity of $\text{MPO}(\mathcal{F}^{i-1} \cup \mathcal{G}^i)$ is $O((|\mathcal{F}^{i-1}| + |\mathcal{G}^i|)|\mathcal{G}^i|)$, it may take a relatively long time, especially when there are thousands of *points* in $\mathcal{F}^{i-1}$ and $\mathcal{G}^i$. Thus, we suggest an efficient update mechanism executed on the surrogate Pareto fronts, $\mathcal{F}^i = \text{MPO}(\tilde{\mathcal{F}}^{i-1} \cup \tilde{\mathcal{G}}^i)$, where $\tilde{\mathcal{G}}^i \subset \mathcal{G}^i$ includes at most $J$ (usually setting $J > K$) best *points* selected from $\boldsymbol{f}(\boldsymbol{\pi}) \in \mathcal{G}^i$ according to $g(\boldsymbol{\pi}|s, \boldsymbol{\lambda}^i)$. The complexity of $\text{MPO}(\tilde{\mathcal{F}}^{i-1} \cup \tilde{\mathcal{G}}^i)$ is then reduced to $O((K + J)J)$, which is able to curtail the overall solving time in practice.

### 4.4 Training and inference

Our NHDE can be applied to different decomposition-based DRL methods, e.g., PMOCO [14] and MDRL [15], and the training algorithm is easy to adapt with slight adjustments, where the one for NHDE-P is presented in Algorithm 1. The key points include three aspects. (1) Multiple weights are sampled to train the same instance (Line 6), since the solving processes for those subproblems are dependent. (2) The HV indicator is adopted in the reward (Line 10). (3) Multiple Pareto-optimal solutions are preserved via MPO (Line 15). Note that when training with a batch, the *point* sets with different sizes of the instances are padded with repetitive reference points, which are masked in the attention, based on the maximum size. The training algorithm of NHDE-M is given in Appendix E.

In the inference phase, for $N$ given weights and diversity factors, the well-trained model is used to sequentially solve $N$ corresponding subproblems, as shown in Figure 1. Moreover, instance augmentation [14] can be also brought into our MPO for each subproblem, i.e., the *points* of all sampled solutions from an instance and its augmented instances are included in $\mathcal{G}^i$. Our NHDE can achieve desirable performance by using only parts of instance augmentation (see Appendix D), since it can already deliver more diverse solutions.

# 5 Experiments

**Problems.** We evaluate the proposed NHDE on three typical MOCO problems that are commonly studied in the neural MOCO literature [13–15], namely the multi-objective traveling salesman problem (MOTSP) [51], multi-objective capacitated vehicle routing problem (MOCVRP) [3], and multi-objective knapsack problem (MOKP) [52]. More specifically, we solve bi-objective TSP (Bi-TSP), tri-objective TSP (Tri-TSP), the bi-objective CVRP (Bi-CVRP) and the bi-objective KP (Bi-KP). For the $M$-objective TSP with $n$ nodes, each node has $M$ sets of 2-dimensional coordinates, where the $m$-th objective value of the solution is calculated with respect to the $m$-th coordinates. Bi-CVRP consists of $n$ customer nodes and a depot node, with each node featured by a 2-dimensional coordinate and each customer node associated with a demand. Following the literature, we consider two conflicting objectives in Bi-CVRP, i.e., the total tour length and the makespan (that is the length of the longest route). Bi-KP is defined by $n$ items, with each taking a weight and two separate values. The $m$-th objective is to maximize the sum of the $m$-th values but not exceed the capacity. Three sizes of these problems are considered, i.e., $n$=20/50/100 for MOTSP and MOCVRP, and $n$=50/100/200 for MOKP. The coordinates, demands, and values are uniformly sampled from $[0,1]^2$, $\{1,\ldots,9\}$, and $[0,1]$, respectively. The vehicle capacity is set to 30/40/50 for MOCVRP20/50/100. The knapsack capacity is set to 12.5/25/25 for MOKP50/100/200.

**Hyperparameters.** We train NHDE-P with 200 epochs, each containing 5,000 randomly generated instances. We use batch size $B=64$ and the Adam [53] optimizer with learning rate $10^{-4}$ ($10^{-5}$ for MOKP) and weight decay $10^{-6}$. During training, $N'=20$ weights are sampled for each instance. During inference, we generate $N=40$ and $N=210$ uniformly distributed weights for $M=2$ and $M=3$, respectively, which are then shuffled so as to counteract biases. The diversity factors linearly shift through the $N$ subproblems from (1,0) to (0,1), which implies a gradual focus from achieving convergence (scalar objective) with a few solutions to ensuring comprehensive performance with a multitude of solutions. We set $K=20$ and $J=200$. See Appendix F for the settings of NHDE-M.

**Baselines.** We compare NHDE with two classes of state-of-the-art methods. (1) The neural methods, including **PMOCO** [14], **MDRL** [15], and DRL-based multiobjective optimization algorithm (**DRL-MOA**) [13], all with POMO as the backbone for single-objective CO subproblems. Our NHDE-P and NHDE-M each train a unified model with the same gradient steps as PMOCO and MDRL, respectively, while DRL-MOA trains 101 (105) models for $M=2$ ($M=3$) with more gradient steps, i.e., the first model with 200 epochs and the remaining models with 5-epoch per model via parameter transfer. (2) The non-learnable methods, including the state-of-the-art MOEA and strong heuristics. Particularly, **PPLS/D-C** [23] is a specialized MOEA for MOCO with local search techniques, including a 2-opt heuristic for MOTSP and MOCVRP, and a greedy transformation heuristic [52] for MOKP, implemented in Python. In addition, LKH [54, 55] and dynamic programming (DP), are employed to solve the weighted-sum (WS) based subproblems for MOTSP and MOKP, denoted as **WS-LKH** and **WS-DP**, respectively. All the methods use WS scalarization for fair comparisons. All the methods are tested with an RTX 3090 GPU and an Intel Xeon 4216 CPU. Our code is publicly available[3].

**Metrics.** We use hypervolume (HV) and the number of non-dominated solutions (|NDS|). A higher HV means better overall performance in terms of convergence and diversity, while |NDS| reflects the diversity when HVs are close. The average HV, gaps with respect to NHDE, and total running time for 200 random test instances are reported. The best (second-best) and its statistically insignificant results at 1% significance level of a Wilcoxon rank-sum test are highlighted in **bold** (underline).

## 5.1 Main results

All results of NHDE-P and the baselines are displayed in Table 1. Given the same number of weights (wt.), NHDE-P significantly surpasses PMOCO for all problems and sizes in terms of HV and |NDS|, which indicates that NHDE-P has the potential to discover diverse and high-quality solutions. When instance augmentation (aug.) is equipped, NHDE-P achieves the smallest gap among the methods in most cases, except Bi-TSP100 and Bi-CVRP100 where WS-LKH and DRL-MOA perform better. However, WS-LKH consumes much longer runtime than NHDE-P due to iterative search (2.7 hours vs 5.6 minutes), and DRL-MOA costs much more training overhead to prepare multiple models for respective weights. Besides, another reason why NHDE-P is inferior to DRL-MOA on Bi-CVRP100 might be that the hypernetwork (inherited from PMOCO) could be hard to cope with the objectives

---

[3]https://github.com/bill-cjb/NHDE

Table 1: Results of NHDE-P on 200 random instances for MOCO problems.

| Method | Bi-TSP20 | | | | Bi-TSP50 | | | | Bi-TSP100 | | | |
|---|---|---|---|---|---|---|---|---|---|---|---|---|
| | HV↑ | \|NDS\|↑ | Gap↓ | Time | HV↑ | \|NDS\|↑ | Gap↓ | Time | HV↑ | \|NDS\|↑ | Gap↓ | Time |
| WS-LKH (40 wt.) | 0.6266 | 14 | 0.46% | 4.1m | 0.6402 | 29 | 0.42% | 42m | **0.7072** | 37 | **-0.31%** | 2.7h |
| PPLS/D-C (200 iter.) | 0.6256 | 71 | 0.62% | 26m | 0.6282 | 213 | 2.29% | 2.8h | 0.6844 | 373 | 2.92% | 11h |
| DRL-MOA (101 models) | 0.6257 | 23 | 0.60% | 6s | 0.6360 | 57 | 1.07% | 9s | 0.6970 | 70 | 1.13% | 21s |
| PMOCO (40 wt.) | 0.6258 | 17 | 0.59% | 4s | 0.6331 | 31 | 1.52% | 5s | 0.6938 | 36 | 1.59% | 8s |
| PMOCO (600 wt.) | 0.6267 | 23 | 0.44% | 27s | 0.6361 | 68 | 1.06% | 53s | 0.6978 | 131 | 1.02% | 2.1m |
| NHDE-P (40 wt.) | 0.6286 | 56 | 0.14% | 19s | 0.6388 | 127 | 0.64% | 53s | 0.7005 | 193 | 0.64% | 1.9m |
| PMOCO (40 wt. aug.) | 0.6266 | 17 | 0.46% | 23s | 0.6377 | 32 | 0.81% | 1.6m | 0.6993 | 37 | 0.81% | 3.0m |
| PMOCO (100 wt. aug.) | 0.6270 | 20 | 0.40% | 1.4m | 0.6395 | 53 | 0.53% | 3.8m | 0.7016 | 76 | 0.48% | 15m |
| NHDE-P (40 wt. aug.) | **0.6295** | 81 | **0.00%** | 1.5m | **0.6429** | 269 | **0.00%** | 2.5m | 0.7050 | 343 | 0.00% | 5.6m |

| Method | Bi-CVRP20 | | | | Bi-CVRP50 | | | | Bi-CVRP100 | | | |
|---|---|---|---|---|---|---|---|---|---|---|---|---|
| | HV↑ | \|NDS\|↑ | Gap↓ | Time | HV↑ | \|NDS\|↑ | Gap↓ | Time | HV↑ | \|NDS\|↑ | Gap↓ | Time |
| PPLS/D-C (200 iter.) | 0.4283 | 14 | 0.46% | 1.3h | 0.4007 | 17 | 2.15% | 9.7h | 0.3946 | 20 | 1.13% | 38h |
| DRL-MOA (101 models) | 0.4287 | 7 | 0.37% | 10s | 0.4076 | 10 | 0.46% | 12s | **0.4055** | 12 | **-1.60%** | 33s |
| PMOCO (40 wt.) | 0.4266 | 6 | 0.86% | 4s | 0.4035 | 7 | 1.47% | 7s | 0.3912 | 6 | 1.98% | 12s |
| PMOCO (300 wt.) | 0.4268 | 7 | 0.81% | 20s | 0.4039 | 9 | 1.37% | 35s | 0.3914 | 8 | 1.93% | 1.2m |
| NHDE-P (40 wt.) | 0.4284 | 12 | 0.44% | 18s | 0.4062 | 14 | 0.81% | 36s | 0.3933 | 10 | 1.45% | 1.1m |
| PMOCO (40 wt. aug.) | 0.4292 | 6 | 0.26% | 8s | 0.4078 | 7 | 0.42% | 15s | 0.3968 | 7 | 0.58% | 1.1m |
| PMOCO (300 wt. aug.) | 0.4294 | 9 | 0.21% | 1.0m | 0.4081 | 10 | 0.34% | 1.8m | 0.3969 | 9 | 0.55% | 7.0m |
| NHDE-P (40 wt. aug.) | **0.4303** | 21 | **0.00%** | 1.2m | **0.4095** | 22 | **0.00%** | 1.5m | 0.3991 | 16 | 0.00% | 2.4m |

| Method | Bi-KP50 | | | | Bi-KP100 | | | | Bi-KP200 | | | |
|---|---|---|---|---|---|---|---|---|---|---|---|---|
| | HV↑ | \|NDS\|↑ | Gap↓ | Time | HV↑ | \|NDS\|↑ | Gap↓ | Time | HV↑ | \|NDS\|↑ | Gap↓ | Time |
| WS-DP (40 wt.) | 0.3560 | 10 | 0.11% | 9.6m | 0.4529 | 16 | 0.26% | 1.3h | 0.3598 | 23 | 0.39% | 3.8h |
| PPLS/D-C (200 iter.) | 0.3528 | 13 | 1.01% | 18m | 0.4480 | 19 | 1.34% | 47m | 0.3541 | 20 | 1.97% | 1.5h |
| DRL-MOA (101 models) | 0.3559 | 21 | 0.14% | 9s | 0.4531 | 38 | 0.22% | 18s | 0.3601 | 48 | 0.30% | 1.0m |
| PMOCO (40 wt.) | 0.3550 | 14 | 0.39% | 6s | 0.4518 | 22 | 0.51% | 9s | 0.3590 | 28 | 0.61% | 25s |
| PMOCO (300 wt.) | 0.3552 | 17 | 0.34% | 29s | 0.4524 | 31 | 0.37% | 1.1m | 0.3597 | 46 | 0.42% | 3.0m |
| NHDE-P (40 wt.) | **0.3564** | 30 | **0.00%** | 29s | **0.4541** | 83 | **0.00%** | 1.0m | **0.3612** | 243 | **0.00%** | 2.7m |

| Method | Tri-TSP20 | | | | Tri-TSP50 | | | | Tri-TSP100 | | | |
|---|---|---|---|---|---|---|---|---|---|---|---|---|
| | HV↑ | \|NDS\|↑ | Gap↓ | Time | HV↑ | \|NDS\|↑ | Gap↓ | Time | HV↑ | \|NDS\|↑ | Gap↓ | Time |
| WS-LKH (210 wt.) | 0.4727 | 78 | 0.82% | 23m | 0.4501 | 189 | 1.90% | 3.5h | 0.5165 | 209 | 0.84% | 12h |
| PPLS/D-C (200 iter.) | 0.4698 | 876 | 1.43% | 1.4h | 0.4174 | 3727 | 9.02% | 3.9h | 0.4376 | 8105 | 15.99% | 14h |
| DRL-MOA (105 models) | 0.4675 | 72 | 1.91% | 5s | 0.4285 | 98 | 6.61% | 9s | 0.4850 | 101 | 6.89% | 19s |
| PMOCO (210 wt.) | 0.4714 | 113 | 1.09% | 11s | 0.4381 | 198 | 4.51% | 18s | 0.4946 | 207 | 5.05% | 39s |
| PMOCO (3003 wt.) | 0.4741 | 264 | 0.52% | 2.3m | 0.4484 | 1339 | 2.27% | 4.6m | 0.5087 | 2330 | 2.34% | 10m |
| NHDE-P (210 wt.) | 0.4758 | 675 | 0.17% | 1.2m | 0.4506 | 2547 | 1.79% | 4.4m | 0.5111 | 4984 | 1.88% | 10m |
| PMOCO (210 wt. aug.) | 0.4727 | 104 | 0.82% | 21m | 0.4471 | 201 | 2.55% | 1.1h | 0.5044 | 209 | 3.17% | 4.2h |
| PMOCO (153 wt. aug.) | 0.4722 | 89 | 0.92% | 15m | 0.4447 | 150 | 3.07% | 47m | 0.5009 | 153 | 3.84% | 3.1h |
| NHDE-P (210 wt. aug.) | **0.4766** | 527 | **0.00%** | 14m | **0.4588** | 9047 | **0.00%** | 30m | **0.5209** | 16999 | **0.00%** | 1.5h |

with imbalanced scales. Considering this drawback of PMOCO, we also apply our method to MDRL, i.e., NHDE-M, and demonstrate that NHDE-M outperforms DRL-MOA on Bi-CVRP100 in Table 2. Also, NHDE-M outperforms MDRL in all cases. More results of NHDE-M are given in Appendix G.

Regarding the inference efficiency, NHDE-P generally takes (tolerably) more runtime than the other learning based methods. To be fair, we further enhance the state-of-the-art PMOCO by adjusting the number of the weights, so that it takes similar or even more runtime compared with NHDE-P. The results in the last two lines for each problem shows that NHDE-P still attains the smaller gaps. We also observe that |NDS| hardly grows along with the increase of number of weights for PMOCO, since numerous solutions to different subproblems could be repetitive. In contrast, NHDE-P is able to produce more diverse solutions with much fewer weights.

## 5.2 Generalization study

To assess the generalization capability of NHDE-P, we compare the trained models (from neural methods) for Bi-TSP100 and the other baselines on 200 random Bi-TSP instances with larger sizes, i.e., Bi-TSP150/200. Three commonly used benchmark instances developed from TSPLIB [56],

Table 2: Results of NHDE-M on 200 random instances for MOCO problems.

| Method | Bi-TSP20 | | | | Bi-TSP50 | | | | Bi-TSP100 | | | |
|---|---|---|---|---|---|---|---|---|---|---|---|---|
| | HV↑ | \|NDS\|↑ | Gap↓ | Time | HV↑ | \|NDS\|↑ | Gap↓ | Time | HV↑ | \|NDS\|↑ | Gap↓ | Time |
| WS-LKH (40 wt.) | 0.6266 | 14 | 0.46% | 4.1m | 0.6402 | 29 | 0.42% | 42m | **0.7072** | 37 | **-0.33%** | 2.7h |
| PPLS/D-C (200 iter.) | 0.6256 | 71 | 0.62% | 26m | 0.6282 | 213 | 2.29% | 2.8h | 0.6844 | 373 | 2.91% | 11h |
| DRL-MOA (101 models) | 0.6257 | 23 | 0.60% | 6s | 0.6360 | 57 | 1.07% | 9s | 0.6970 | 70 | 1.12% | 21s |
| MDRL (40 wt.) | 0.6264 | 20 | 0.49% | 2s | 0.6342 | 33 | 1.35% | 3s | 0.6940 | 36 | 1.55% | 8s |
| NHDE-M (40 wt.) | 0.6287 | 58 | 0.13% | 20s | 0.6393 | 132 | 0.56% | 57s | 0.7008 | 195 | 0.58% | 2.0m |
| MDRL (40 wt. aug.) | 0.6267 | 18 | 0.44% | 21s | 0.6384 | 34 | 0.70% | 1.5m | 0.6995 | 38 | 0.77% | 3.3m |
| NHDE-M (40 wt. aug.) | **0.6295** | 81 | **0.00%** | 1.5m | **0.6429** | 273 | **0.00%** | 2.6m | 0.7049 | 339 | 0.00% | 5.5m |

| Method | Bi-CVRP20 | | | | Bi-CVRP50 | | | | Bi-CVRP100 | | | |
|---|---|---|---|---|---|---|---|---|---|---|---|---|
| | HV↑ | \|NDS\|↑ | Gap↓ | Time | HV↑ | \|NDS\|↑ | Gap↓ | Time | HV↑ | \|NDS\|↑ | Gap↓ | Time |
| PPLS/D-C (200 iter.) | 0.4287 | 15 | 0.42% | 1.6h | 0.4007 | 17 | 2.34% | 9.7h | 0.3946 | 20 | 3.14% | 38h |
| DRL-MOA (101 models) | 0.4287 | 7 | 0.42% | 10s | 0.4076 | 10 | 0.66% | 12s | 0.4055 | 12 | 0.47% | 33s |
| MDRL (40 wt.) | 0.4284 | 9 | 0.49% | 3s | 0.4057 | 5 | 1.12% | 5s | 0.4015 | 0 | 1.45% | 10s |
| NHDE-M (40 wt.) | 0.4296 | 16 | 0.21% | 23s | 0.4086 | 20 | 0.41% | 47s | 0.4053 | 18 | 0.52% | 1.4m |
| MDRL (40 wt. aug.) | 0.4293 | 9 | 0.28% | 5s | 0.4073 | 11 | 0.73% | 16s | 0.4040 | 11 | 0.83% | 1.0m |
| NHDE-M (40 wt. aug.) | **0.4305** | 24 | **0.00%** | 1.2m | **0.4103** | 29 | **0.00%** | 1.6m | **0.4074** | 26 | **0.00%** | 2.7m |

| Method | Bi-KP50 | | | | Bi-KP100 | | | | Bi-KP200 | | | |
|---|---|---|---|---|---|---|---|---|---|---|---|---|
| | HV↑ | \|NDS\|↑ | Gap↓ | Time | HV↑ | \|NDS\|↑ | Gap↓ | Time | HV↑ | \|NDS\|↑ | Gap↓ | Time |
| WS-DP (40 wt.) | 0.3560 | 10 | 0.17% | 9.6m | 0.4529 | 16 | 0.29% | 1.3h | 0.3598 | 23 | 0.30% | 3.8h |
| PPLS/D-C (200 iter.) | 0.3528 | 13 | 1.07% | 18m | 0.4480 | 19 | 1.37% | 47m | 0.3541 | 20 | 1.88% | 1.5h |
| DRL-MOA (101 models) | 0.3559 | 21 | 0.20% | 9s | 0.4531 | 38 | 0.24% | 18s | 0.3601 | 48 | 0.22% | 1.0m |
| MDRL (40 wt.) | 0.3559 | 17 | 0.20% | 4s | 0.4528 | 25 | 0.31% | 8s | 0.3594 | 31 | 0.42% | 24s |
| NHDE-M (40 wt.) | **0.3566** | 41 | **0.00%** | 31s | **0.4542** | 93 | **0.00%** | 1.0m | **0.3609** | 160 | **0.00%** | 2.8m |

| Method | Tri-TSP20 | | | | Tri-TSP50 | | | | Tri-TSP100 | | | |
|---|---|---|---|---|---|---|---|---|---|---|---|---|
| | HV↑ | \|NDS\|↑ | Gap↓ | Time | HV↑ | \|NDS\|↑ | Gap↓ | Time | HV↑ | \|NDS\|↑ | Gap↓ | Time |
| WS-LKH (210 wt.) | 0.4727 | 78 | 0.82% | 23m | 0.4501 | 189 | 2.00% | 3.5h | **0.5165** | 209 | **-0.92%** | 12h |
| PPLS/D-C (200 iter.) | 0.4698 | 876 | 1.43% | 1.4h | 0.4174 | 3727 | 9.12% | 3.9h | 0.4376 | 8105 | 14.50% | 14h |
| DRL-MOA (105 models) | 0.4675 | 72 | 1.91% | 5s | 0.4285 | 98 | 6.71% | 9s | 0.4850 | 101 | 5.24% | 19s |
| MDRL (210 wt.) | 0.4723 | 126 | 0.90% | 14s | 0.4388 | 199 | 4.46% | 20s | 0.4956 | 207 | 3.17% | 40s |
| NHDE-M (210 wt.) | 0.4763 | 783 | 0.06% | 1.4m | 0.4512 | 2636 | 1.76% | 4.7m | 0.4997 | 4056 | 2.36% | 11m |
| MDRL (210 wt. aug.) | 0.4727 | 107 | 0.82% | 14m | 0.4473 | 202 | 2.61% | 53m | 0.5056 | 209 | 1.21% | 4.3h |
| NHDE-M (210 wt. aug.) | **0.4766** | 748 | **0.00%** | 13m | **0.4593** | 10850 | **0.00%** | 30m | 0.5118 | 13216 | 0.00% | 1.5h |

Table 3: Results on 200 random instances for larger-scale problems.

| Method | Bi-TSP150 | | | | Bi-TSP200 | | | |
|---|---|---|---|---|---|---|---|---|
| | HV↑ | \|NDS\|↑ | Gap↓ | Time | HV↑ | \|NDS\|↑ | Gap↓ | Time |
| WS-LKH (40 wt.) | **0.7075** | 39 | **-0.90%** | 5.3h | **0.7435** | 40 | **-1.52%** | 8.5h |
| PPLS/D-C (200 iter.) | 0.6784 | 473 | 3.25% | 21h | 0.7106 | 512 | 2.98% | 32h |
| DRL-MOA (101 models) | 0.6901 | 73 | 1.58% | 45s | 0.7219 | 75 | 1.43% | 87s |
| PMOCO (40 wt.) | 0.6891 | 37 | 1.73% | 22s | 0.7215 | 38 | 1.49% | 41s |
| PMOCO (400 wt.) | 0.6938 | 160 | 1.06% | 3.7m | 0.7259 | 186 | 0.89% | 6.8m |
| NHDE-P (40 wt.) | 0.6964 | 231 | 0.68% | 3.0m | 0.7280 | 259 | 0.60% | 4.3m |
| PMOCO (40 wt. aug.) | 0.6944 | 38 | 0.97% | 20m | 0.7264 | 39 | 0.82% | 40m |
| NHDE-P (40 wt. aug.) | 0.7012 | 372 | 0.00% | 14m | 0.7324 | 384 | 0.00% | 26m |

i.e., KroAB100, KroAB150, and KroAB200, are also tested. The comparison results and Pareto fronts are demonstrated in Table 3 and Figure 3, respectively. As shown, NHDE-P outperforms the state-of-the-art MOEA (i.e., PPLS/D-C) and other neural methods significantly, in terms of HV and |NDS|, which means a superior generalization capability. The figure again verifies that NHDE-P generates a more extended Pareto front than PMOCO, showing a better diversity. Although PPLS/D-C finds a large number of solutions, they are inferior in terms of the optimality, with a biased distribution (e.g., crowded in certain regions). In contrast, NHDE-P generates a more well-distributed Pareto front with stronger convergence. More results on generalization are given in Appendix H.

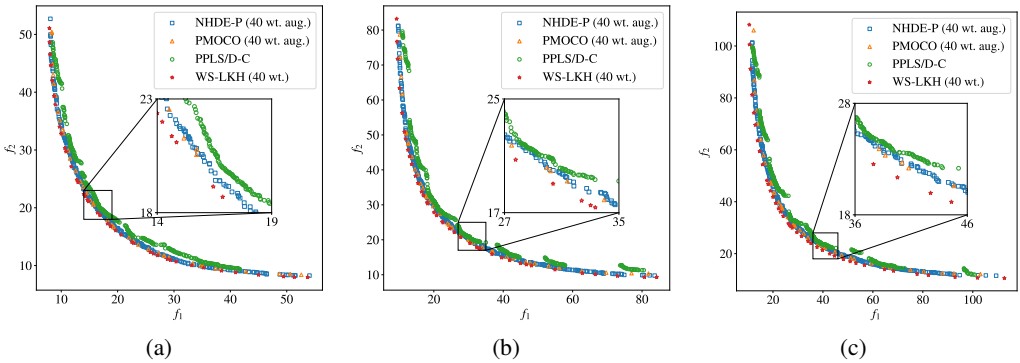

Figure 3: Pareto fronts of benchmark instances. (a) KroAB100. (b) KroAB150. (c) KroAB200.

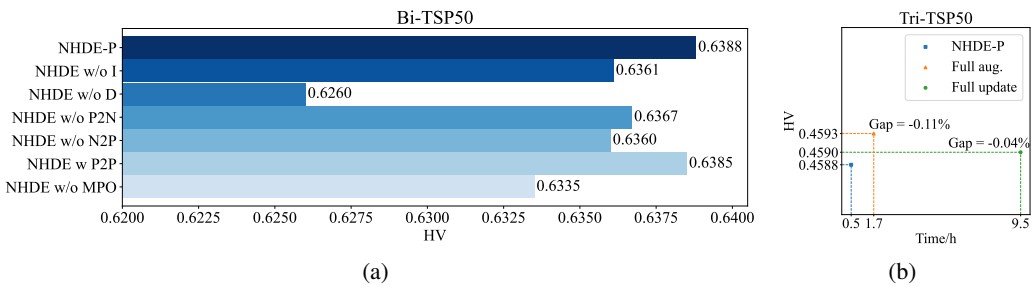

Figure 4: Ablation study. (a) Effects of indicator-enhanced DRL, HGA, and MPO. (b) Effects of the efficient update of MPO and partial instance augmentation.

### 5.3 Ablation Study

To analyze the effect of the indicator-enhanced DRL, we compare NHDE-P with decomposition-based DRL without indicator (NHDE w/o I) and indicator-based DRL without decomposition (NHDE w/o D). To verify the valid design of HGA, three attention-based variants, which are formed by removing *point-to-node* attention (NHDE w/o P2N), removing *node-to-point* attention (NHDE w/o N2P), and adding *point-to-point* attention (NHDE w P2P), are involved for comparison. To assess the impact of MPO, NHDE w/o MPO is also evaluated. More details about these variants are presented in Appendix I. As seen from Figure 4(a). The performance of NHDE-P is significantly impaired when any of the components is ablated. Instead, NHDE w P2P degrades a bit, which reveals that the extra *point-to-point* attention may bring noises into the model. Moreover, we evaluate the effectiveness of the efficient update of MPO and partial instance augmentation in Figure 4(b), which shows that either of them saliently diminishes solving time with only a little sacrifice of the performance.

## 6 Conclusion

This paper proposes a novel NHDE for MOCO problems. NHDE impedes repetitive solutions from different subproblems via indicator-enhanced DRL with a HGA model, and digs more solutions in the neighborhood of each subproblem with an MPO strategy. Our generic NHDE can be deployed to different neural MOCO methods. The experimental results on three classic MOCO problems showed the superiority of NHDE, especially with regard to the diversity of Pareto set. A limitation is that the HV calculation would expend additional computational time, which might hinder the scalability of NHDE for solving much larger problems with many objectives. In the future, we will explore alternative schemes like the HV approximations [57, 58] to further promote the training efficiency of NHDE, and we also intend to apply it to tackle real-world MOCO problems.

## Acknowledgments and disclosure of funding

This work is supported by the National Natural Science Foundation of China (62072483), and the Guangdong Basic and Applied Basic Research Foundation (2022A1515011690, 2021A1515012298).

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
