# Neural Multi-Objective Combinatorial Optimization with Diversity Enhancement (Appendix)

## A  Reference point and hypervolume ratio

The normalized hypervolume (HV) ratio is calculated as $\mathrm{HV}'_{\boldsymbol{r}}(\mathcal{F}) = \mathrm{HV}_{\boldsymbol{r}}(\mathcal{F})/\prod_{i=1}^{M}|r_i - z_i|$, where $\boldsymbol{r}$ is a reference point satisfying $r_i > \max\{f_i(\boldsymbol{x})|\boldsymbol{f}(\boldsymbol{x}) \in \mathcal{F}\}$ and $\boldsymbol{z}$ is an ideal point satisfying $z_i < \min\{f_i(\boldsymbol{x})|\boldsymbol{f}(\boldsymbol{x}) \in \mathcal{F}\}^4$, $\forall i \in \{1, \ldots, M\}$. The used $\boldsymbol{r}$ and $\boldsymbol{z}$ are given in Table 4.

Table 4: Reference points and ideal points

| Problem | Size | $\boldsymbol{r}$ | $\boldsymbol{z}$ |
|---------|------|------------------|------------------|
| Bi-TSP  | 20   | (20, 20)         | (0, 0)           |
|         | 50   | (35, 35)         | (0, 0)           |
|         | 100  | (65, 65)         | (0, 0)           |
|         | 150  | (85, 85)         | (0, 0)           |
|         | 200  | (115, 115)       | (0, 0)           |
| Bi-CVRP | 20   | (30, 4)          | (0, 0)           |
|         | 50   | (45, 4)          | (0, 0)           |
|         | 100  | (80, 4)          | (0, 0)           |
| Bi-KP   | 50   | (5, 5)           | (30, 30)         |
|         | 100  | (20, 20)         | (50, 50)         |
|         | 200  | (30, 30)         | (75, 75)         |
| Tri-TSP | 20   | (20, 20, 20)     | (0, 0)           |
|         | 50   | (35, 35, 35)     | (0, 0)           |
|         | 100  | (65, 65, 65)     | (0, 0)           |

## B  Details of NHDE-P and NHDE-M

NHDE-P, deploying NHDE to PMOCO [14], employs a hypernetwork to tackle the weight $\boldsymbol{\lambda}$ and diversity factor $\boldsymbol{w}$ for the corresponding subproblem. Specifically, according to the given $\boldsymbol{\lambda}$ and $\boldsymbol{w}$, the hypernetwork generates the decoder parameters of the heterogeneous graph attention (HGA) model $\boldsymbol{\theta}$, which is an encoder-decoder-styled architecture, i.e., $\boldsymbol{\theta}(\boldsymbol{\lambda}, \boldsymbol{w}) = [\boldsymbol{\theta}_{\mathrm{en}}, \boldsymbol{\theta}_{\mathrm{de}}(\boldsymbol{\lambda}, \boldsymbol{w})]$, as shown in Figure 5. Following [14], the hypernetwork adopts a simple MLP model with two 256-dimensional hidden layers and ReLu activation. The MLP first maps an input with $M + 2$ dimensions to a hidden embedding $\boldsymbol{h}(\boldsymbol{\lambda}, \boldsymbol{w})$, which is then used to generate the decoder parameters by linear projection.

NHDE-M, deploying NHDE to MDRL [15], consists of three processes. In the meta-learning process, a meta-model $\boldsymbol{\theta}_{\mathrm{meta}}$, whose architecture is the same as the HGA model $\boldsymbol{\theta}$, is trained by sampling tasks from the whole task space. In the fine-tuning process, according to the given $\boldsymbol{\lambda}$ and $\boldsymbol{w}$, $\boldsymbol{\theta}_{\mathrm{meta}}$ is then fine-tuned using fine-tuning instances with a few gradient steps to derive the corresponding submodel. In the inference process, the submodel is used to solve the corresponding subproblem.

## C  Node features and context embedding

The input dimensions of the node features vary with different problems. The inputs of the $M$-objective TSP are $n$ nodes with $2M$-dimensional features. The inputs of Bi-CVRP are $n$ customer nodes with 3-dimensional features and a depot node with 2-dimensional features. The inputs of Bi-KP are $n$ nodes with 3-dimensional features.

---

[4]$r_i < \min\{f_i(\boldsymbol{x})|\boldsymbol{f}(\boldsymbol{x}) \in \mathcal{F}\}$ and $z_i > \max\{f_i(\boldsymbol{x})|\boldsymbol{f}(\boldsymbol{x}) \in \mathcal{F}\}$ for maximization problems, e.g., Bi-KP.

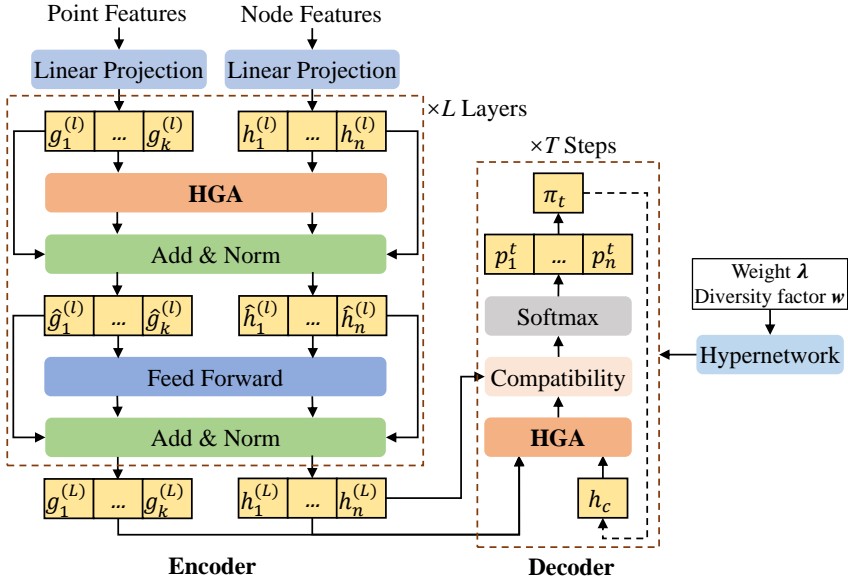

Figure 5: Hypernetwork-based heterogeneous graph attention (HGA) model of NHDE-P.

At step $t$ in the decoder, a context embedding $\boldsymbol{h}_c$ is used to calculated the probability of node selection. For MOTSP, $\boldsymbol{h}_c$ is defined as the concatenation of the graph embedding $\bar{\boldsymbol{h}} = \sum_{u=1}^{n} \boldsymbol{h}_u / n$, the embedding of the first node $\boldsymbol{h}_{\boldsymbol{\pi}_1}$, and the embedding of the last node $\boldsymbol{h}_{\boldsymbol{\pi}_{t-1}}$. For MOCVRP, $\boldsymbol{h}_c$ is defined as the concatenation of the graph embedding $\bar{\boldsymbol{h}}$, the embedding of the last node $\boldsymbol{h}_{\boldsymbol{\pi}_{t-1}}$, and the remaining vehicle capacity. For MOKP, $\boldsymbol{h}_c$ is defined as the concatenation of the graph embedding $\bar{\boldsymbol{h}}$ and the remaining knapsack capacity.

A masking mechanism is adopted in each decoding step to ensure the solution feasibility. For MOTSP, the visited nodes are masked. For MOCVRP (MOKP), besides the visited nodes, those with a demand (weight) larger than the remaining vehicle (knapsack) capacity are also masked.

## D Instance augmentation

In the inference process, an instance can be transformed into other variants sharing the same optimal solutions, so as to augment the performance. An instance of Bi-CVRP has 8 transformations, and an instance of $M$-objective TSP has $8^M$ transformations [14] due to the full transformation permutation of $M$ groups of 2-dimensional coordinates, where each group has 8 transformations [36], $\{(x, y), (1 - x, y), (x, 1 - y), (1 - x, 1 - y), (y, x), (1 - y, x), (y, 1 - x), (1 - y, 1 - x)\}$.

Our NHDE can effectively enhance the performance only using partial instance augmentation, which can reduce the solving time, since it can already achieve high diversity. Specifically, for $M$-objective TSP, NHDE adopts the full permutation of the first 4 transformations and last 4 transformations, respectively, thereby a total of $2 \times 4^M$ transformations.

## E Training and fine-tuning of NHDE-M

The training algorithm of NHDE-M in the meta-learning process, adapted from that of MDRL [15], is outlined in Algorithm 2. Also, the three key adjustments based on MDRL are captured in Line 7, Line13, and Line 18. In the fine-tuning process, for the given $N$ weights $\boldsymbol{\lambda}$ as well as diversity factors $\boldsymbol{w}$, $N$ submodels are fine-tuned from the well-trained meta-model to solve the MOCO problem. The fine-tuning algorithm is presented in Algorithm 3.

---

**Algorithm 2** Training algorithm of NHDE-M

---

1: **Input:** weight distribution $\Lambda$, diversity-factor distribution $\mathcal{W}$, instance distribution $\mathcal{S}$, initial meta-learning rate $\epsilon_0$, number of meta-iterations $T_m$, number of sampling steps per meta-iteration $N'$, number of sampled weights per sampling step $\tilde{N}$, number of update steps of the submodel $E$, batch size $B$, instance size $n$

2: Initialize the meta-model $\boldsymbol{\theta}$

3: $\epsilon \leftarrow \epsilon_0$

4: **for** $t_m = 1$ to $T_m$ **do**

5: $\quad s_{e,i} \sim$ **SampleInstance**$(\mathcal{S})$  $\forall e \in \{1, \cdots, E\}$  $\forall i \in \{1, \cdots, B\}$

6: $\quad$ Initialize $\mathcal{F}_{e,i} \leftarrow \emptyset$  $\forall e, i$

7: $\quad$ **for** $n' = 1$ to $N'$ **do**

8: $\quad\quad$ **for** $\tilde{n} = 1$ to $\tilde{N}$ **do**

9: $\quad\quad\quad \boldsymbol{\lambda} \sim$ **SampleWeight**$(\Lambda)$

10: $\quad\quad\quad \boldsymbol{w} \sim$ **SampleDiversityFactor**$(\mathcal{W})$

11: $\quad\quad\quad$ **for** $e = 1$ to $E$ **do**

12: $\quad\quad\quad\quad \boldsymbol{\pi}_i^j \sim$ **SampleSolution**$(P_{\boldsymbol{\theta}^{\tilde{n}}}(\cdot|s_{e,i}, \tilde{\mathcal{F}}_{\boldsymbol{r},e,i}))$  $\forall i \in \{1, \cdots, B\}$  $\forall j \in \{1, \cdots, n\}$

13: $\quad\quad\quad\quad R_i^j \leftarrow -w_1 g(\boldsymbol{\pi}_i^j|s_{e,i}, \boldsymbol{\lambda}) + w_2 \text{HV}_{\boldsymbol{r}}(\tilde{\mathcal{F}}_{e,i} \cup \{\boldsymbol{\pi}_i^j\})$  $\forall i, j$

14: $\quad\quad\quad\quad b_i \leftarrow \frac{1}{n}\sum_{j=1}^n(-R_i^j)$  $\forall i$

15: $\quad\quad\quad\quad \nabla\mathcal{J}(\boldsymbol{\theta}^{\tilde{n}}) \leftarrow \frac{1}{Bn}\sum_{i=1}^B\sum_{j=1}^n[(-R_i^j - b_i)\nabla_{\boldsymbol{\theta}^{\tilde{n}}}\log P_{\boldsymbol{\theta}^{\tilde{n}}}(\boldsymbol{\pi}_i^j|s_i, \tilde{\mathcal{F}}_{\boldsymbol{r},i})]$

16: $\quad\quad\quad\quad \boldsymbol{\theta}^{\tilde{n}} \leftarrow$ **Adam**$(\boldsymbol{\theta}^{\tilde{n}}, \nabla\mathcal{J}(\boldsymbol{\theta}^{\tilde{n}}))$

17: $\quad\quad\quad\quad \mathcal{G}_i \leftarrow \{\boldsymbol{\pi}_i^1, \ldots, \boldsymbol{\pi}_i^n\}$  $\forall i$

18: $\quad\quad\quad\quad \mathcal{F}_{e,i} \leftarrow \text{MPO}(\tilde{\mathcal{F}}_{e,i} \cup \tilde{\mathcal{G}}_i)$  $\forall e, i$

19: $\quad\quad\quad$ **end for**

20: $\quad\quad$ **end for**

21: $\quad\quad \boldsymbol{\theta} \leftarrow \boldsymbol{\theta} + \epsilon(\frac{1}{\tilde{N}}\sum_{\tilde{n}=1}^{\tilde{N}}\boldsymbol{\theta}^{\tilde{n}} - \boldsymbol{\theta})$

22: $\quad\quad \epsilon \leftarrow \epsilon - \epsilon_0/(T_m \times N')$

23: $\quad$ **end for**

24: **end for**

25: **Output:** The parameter of the meta-model $\boldsymbol{\theta}$

---

---

**Algorithm 3** Fine-tuning algorithm of NHDE-M

---

1: **Input:** instance distribution $\mathcal{S}$, weights $\boldsymbol{\lambda}^1, \ldots, \boldsymbol{\lambda}^N$, diversity factors $\boldsymbol{w}^1, \ldots, \boldsymbol{w}^N$, number of fine-tuning steps of the submodel $E_f$, batch size $B$, instance size $n$, well-trained meta-model $\boldsymbol{\theta}$

2: $s_{e,i} \sim$ **SampleInstance**$(\mathcal{S})$  $\forall e \in \{1, \cdots, E_f\}$  $\forall i \in \{1, \cdots, B\}$

3: Initialize $\mathcal{F}_{e,i} \leftarrow \emptyset$  $\forall e, i$

4: **for** $\tilde{n} = 1$ to $N$ **do**

5: $\quad \boldsymbol{\theta}^{\tilde{n}} \leftarrow \boldsymbol{\theta}$

6: $\quad$ **for** $e = 1$ to $E$ **do**

7: $\quad\quad \boldsymbol{\pi}_i^j \sim$ **SampleSolution**$(P_{\boldsymbol{\theta}^{\tilde{n}}}(\cdot|s_{e,i}, \tilde{\mathcal{F}}_{\boldsymbol{r},e,i}))$  $\forall i \in \{1, \cdots, B\}$  $\forall j \in \{1, \cdots, n\}$

8: $\quad\quad R_i^j \leftarrow -w_1 g(\boldsymbol{\pi}_i^j|s_{e,i}, \boldsymbol{\lambda}) + w_2 \text{HV}_{\boldsymbol{r}}(\tilde{\mathcal{F}}_{e,i} \cup \{\boldsymbol{\pi}_i^j\})$  $\forall i, j$

9: $\quad\quad b_i \leftarrow \frac{1}{n}\sum_{j=1}^n(-R_i^j)$  $\forall i$

10: $\quad\quad \nabla\mathcal{J}(\boldsymbol{\theta}^{\tilde{n}}) \leftarrow \frac{1}{Bn}\sum_{i=1}^B\sum_{j=1}^n[(-R_i^j - b_i)\nabla_{\boldsymbol{\theta}^{\tilde{n}}}\log P_{\boldsymbol{\theta}^{\tilde{n}}}(\boldsymbol{\pi}_i^j|s_i, \tilde{\mathcal{F}}_{\boldsymbol{r},i})]$

11: $\quad\quad \boldsymbol{\theta}^{\tilde{n}} \leftarrow$ **Adam**$(\boldsymbol{\theta}^{\tilde{n}}, \nabla\mathcal{J}(\boldsymbol{\theta}^{\tilde{n}}))$

12: $\quad\quad \mathcal{G}_i \leftarrow \{\boldsymbol{\pi}_i^1, \ldots, \boldsymbol{\pi}_i^n\}$  $\forall i$

13: $\quad\quad \mathcal{F}_{e,i} \leftarrow \text{MPO}(\tilde{\mathcal{F}}_{e,i} \cup \tilde{\mathcal{G}}_i)$  $\forall e, i$

14: $\quad$ **end for**

15: **end for**

16: **Output:** The parameters of the fine-tuned submodels $\boldsymbol{\theta}^1, \ldots, \boldsymbol{\theta}^N$

---

# F  Hyperparameters of NHDE-M

NHDE-M trains a meta-model with 150 meta-iterations and initial meta-learning rate $\epsilon_0 = 1$. We set $N' = 20$, $\tilde{N} = M$, and $E = 100$. We use batch size $B = 64$ and the Adam [53] optimizer with

Table 5: Results of NHDE-M compared with MDRL with close or more total solving time on 200 random instances of MOCO problems.

| Method | Bi-TSP20 HV↑ | \|NDS\|↑ | Gap↓ | Time | Bi-TSP50 HV↑ | \|NDS\|↑ | Gap↓ | Time | Bi-TSP100 HV↑ | \|NDS\|↑ | Gap↓ | Time |
|---|---|---|---|---|---|---|---|---|---|---|---|---|
| MDRL (40 wt.) | 0.6264 | 20 | 0.49% | 2s | 0.6342 | 33 | 1.35% | 3s | 0.6940 | 36 | 1.55% | 8s |
| MDRL (600 wt.) | 0.6287 | 54 | 0.13% | 29s | 0.6380 | 133 | 0.76% | 64s | 0.7006 | 185 | 0.61% | 2.1m |
| NHDE-M (40 wt.) | 0.6287 | 58 | 0.13% | 20s | 0.6393 | 132 | 0.56% | 57s | 0.7008 | 195 | 0.58% | 2.0m |
| MDRL (40 wt. aug.) | 0.6267 | 18 | 0.44% | 21s | 0.6384 | 34 | 0.70% | 1.5m | 0.6995 | 38 | 0.77% | 3.3m |
| MDRL (100 wt. aug.) | 0.6271 | 23 | 0.38% | 1.2m | 0.6408 | 67 | 0.33% | 3.6m | 0.7023 | 82 | 0.37% | 16m |
| NHDE-M (40 wt. aug.) | **0.6295** | 81 | **0.00%** | 1.5m | **0.6429** | 273 | **0.00%** | 2.6m | **0.7049** | 339 | **0.00%** | 5.5m |

| Method | Bi-CVRP20 HV↑ | \|NDS\|↑ | Gap↓ | Time | Bi-CVRP50 HV↑ | \|NDS\|↑ | Gap↓ | Time | Bi-CVRP100 HV↑ | \|NDS\|↑ | Gap↓ | Time |
|---|---|---|---|---|---|---|---|---|---|---|---|---|
| MDRL (40 wt.) | 0.4284 | 9 | 0.49% | 3s | 0.4057 | 5 | 1.12% | 5s | 0.4015 | 0 | 1.45% | 10s |
| MDRL (300 wt.) | 0.4296 | 17 | 0.21% | 23s | 0.4089 | 21 | 0.34% | 49s | 0.4078 | 21 | -0.10% | 1.5m |
| NHDE-M (40 wt.) | 0.4296 | 16 | 0.21% | 23s | 0.4086 | 20 | 0.41% | 47s | 0.4053 | 18 | 0.52% | 1.4m |
| MDRL (40 wt. aug.) | 0.4293 | 9 | 0.28% | 5s | 0.4073 | 11 | 0.73% | 16s | 0.4040 | 11 | 0.83% | 1.0m |
| MDRL (300 wt. aug.) | 0.4302 | 16 | 0.07% | 1.0m | **0.4103** | 24 | **0.00%** | 2.1m | **0.4086** | 24 | **-0.29%** | 7.7m |
| NHDE-M (40 wt. aug.) | **0.4305** | 24 | **0.00%** | 1.2m | **0.4103** | 29 | **0.00%** | 1.6m | 0.4074 | 26 | 0.00% | 2.7m |

| Method | Bi-KP50 HV↑ | \|NDS\|↑ | Gap↓ | Time | Bi-KP100 HV↑ | \|NDS\|↑ | Gap↓ | Time | Bi-KP200 HV↑ | \|NDS\|↑ | Gap↓ | Time |
|---|---|---|---|---|---|---|---|---|---|---|---|---|
| MDRL (40 wt.) | 0.3559 | 17 | 0.20% | 4s | 0.4528 | 25 | 0.31% | 8s | 0.3594 | 31 | 0.42% | 24s |
| MDRL (300 wt.) | 0.3563 | 29 | 0.08% | 30s | 0.4536 | 58 | 0.13% | 1.0m | 0.3606 | 95 | 0.08% | 3.1m |
| NHDE-M (40 wt.) | **0.3566** | 41 | **0.00%** | 31s | **0.4542** | 93 | **0.00%** | 1.0m | **0.3609** | 160 | **0.00%** | 2.8m |

| Method | Tri-TSP20 HV↑ | \|NDS\|↑ | Gap↓ | Time | Tri-TSP50 HV↑ | \|NDS\|↑ | Gap↓ | Time | Tri-TSP100 HV↑ | \|NDS\|↑ | Gap↓ | Time |
|---|---|---|---|---|---|---|---|---|---|---|---|---|
| MDRL (210 wt.) | 0.4723 | 126 | 0.90% | 14s | 0.4388 | 199 | 4.46% | 20s | 0.4956 | 207 | 3.17% | 40s |
| MDRL (3003 wt.) | 0.4761 | 479 | 0.10% | 2.6m | 0.4512 | 1927 | 1.76% | 5.1m | 0.5104 | 2400 | 0.27% | 10m |
| NHDE-M (210 wt.) | 0.4763 | 783 | 0.06% | 1.4m | 0.4512 | 2636 | 1.76% | 4.7m | 0.4997 | 4056 | 2.36% | 11m |
| MDRL (210 wt. aug.) | 0.4727 | 107 | 0.82% | 14m | 0.4473 | 202 | 2.61% | 53m | 0.5056 | 209 | 1.21% | 4.3h |
| MDRL (153 wt. aug.) | 0.4721 | 92 | 0.94% | 11m | 0.4448 | 150 | 3.16% | 39m | 0.5022 | 153 | 1.88% | 3.2h |
| NHDE-M (210 wt. aug.) | **0.4766** | 748 | **0.00%** | 13m | **0.4593** | 10850 | **0.00%** | 30m | **0.5118** | 13216 | **0.00%** | 1.5h |

learning rate $10^{-4}$. During fine-tuning and inference, the number of fine-tuning steps $E_f$ is set to 50, and Adam with learning rate $10^{-4}$ is used. $N=40$ and $N=210$ uniformly distributed weights are generated and then shuffled for $M=2$ and $M=3$, respectively. $N$ diversity factors linearly change from (1,0) to (0,1). For the compared MDRL, the settings are the same as NHDE-M except 3000 meta-iterations are used, so that MDRL and NHDE-M execute the same number of gradient steps.

## G  Experimental results of NHDE-M

### G.1  More results of NHDE-M

NHDE-M usually spends relatively more inference time than MDRL with the same number of weights. Hence, we adjust the number of weights of MDRL, making its inference time close to or longer than NHDE-M for fair comparisons, as shown in Table 5. Nonetheless, NHDE-M is still superior to MDRL in most cases. Without instance augmentation (aug.), NHDE-M is inferior to MDRL on Bi-CVRP50, Bi-CVRP100, and Tri-TSP100. Instance augmentation can effectively boost the performance of NHDE-M, where NHDE-M (aug.) is only inferior to MDRL (aug.) on Bi-CVRP100. Note that, for a given weight, MDRL needs to further fine-tune the meta-model with $E_f = 50$ gradient steps to derive a submodel to the corresponding subproblem, which means that the increasing number of weights would cause considerable extra fine-tuning costs.

### G.2  Generalization study of NHDE-M

We assess the zero-shot generalization capability of NHDE-M, which is trained and fine-tuned on Bi-TSP100, and tested on 200 random larger-scale Bi-TSP instances, i.e., Bi-TSP150/200, and three commonly used benchmark instances, i.e., KroAB100/150/200. The results are gathered in Tables 6 and 7. The Pareto fronts of the benchmark instances obtained by various methods are also visualized

Table 6: Results of NHDE-M on 200 random instances of larger-scale problems.

| Method | Bi-TSP150 | | | | Bi-TSP200 | | | |
| --- | --- | --- | --- | --- | --- | --- | --- | --- |
| | HV↑ | \|NDS\|↑ | Gap↓ | Time | HV↑ | \|NDS\|↑ | Gap↓ | Time |
| WS-LKH (40 wt.) | **0.7075** | 39 | **-0.90%** | 5.3h | **0.7435** | 40 | **-1.38%** | 8.5h |
| PPLS/D-C (200 iter.) | 0.6784 | 473 | 3.25% | 21h | 0.7106 | 512 | 3.11% | 32h |
| DRL-MOA (101 models) | 0.6901 | 73 | 1.58% | 45s | 0.7219 | 75 | 1.57% | 87s |
| MDRL (40 wt.) | 0.6894 | 37 | 1.68% | 23s | 0.7227 | 38 | 1.46% | 42s |
| MDRL (400 wt.) | 0.6958 | 176 | 0.77% | 3.7m | 0.7284 | 190 | 0.68% | 6.9m |
| NHDE-M (40 wt.) | 0.6970 | 239 | 0.60% | 3.0m | 0.7297 | 268 | 0.50% | 4.1m |
| MDRL (40 wt. aug.) | 0.6948 | 39 | 0.91% | 21m | 0.7275 | 39 | 0.80% | 41m |
| NHDE-M (40 wt. aug.) | 0.7012 | 373 | 0.00% | 14m | 0.7334 | 395 | 0.00% | 25m |

Table 7: Results of NHDE-M on benchmark instances.

| Method | KroAB100 | | | | KroAB150 | | | | KroAB200 | | | |
| --- | --- | --- | --- | --- | --- | --- | --- | --- | --- | --- | --- | --- |
| | HV↑ | \|NDS\|↑ | Gap↓ | Time | HV↑ | \|NDS\|↑ | Gap↓ | Time | HV↑ | \|NDS\|↑ | Gap↓ | Time |
| WS-LKH (40 wt.) | **0.7007** | 40 | **-0.42%** | 53s | **0.6989** | 39 | **-0.92%** | 1.9m | **0.7404** | 40 | **-1.48%** | 2.2m |
| PPLS/D-C (200 iter.) | 0.6785 | 388 | 2.77% | 31m | 0.6659 | 441 | 3.84% | 1.1h | 0.7100 | 491 | 2.69% | 3.1h |
| DRL-MOA (101 models) | 0.6903 | 67 | 1.07% | 10s | 0.6794 | 72 | 1.89% | 18s | 0.7185 | 73 | 1.52% | 23s |
| MDRL (40 wt.) | 0.6869 | 37 | 1.56% | 5s | 0.6810 | 36 | 1.66% | 9s | 0.7184 | 39 | 1.54% | 12s |
| NHDE-M (40 wt.) | 0.6940 | 183 | 0.54% | 7s | 0.6879 | 232 | 0.66% | 12s | 0.7253 | 275 | 0.59% | 16s |
| MDRL (40 wt. aug.) | 0.6928 | 37 | 0.72% | 7s | 0.6857 | 38 | 0.98% | 10s | 0.7234 | 40 | 0.85% | 15s |
| NHDE-M (40 wt. aug.) | 0.6978 | 341 | 0.00% | 10s | 0.6925 | 370 | 0.00% | 12s | 0.7296 | 393 | 0.00% | 16s |

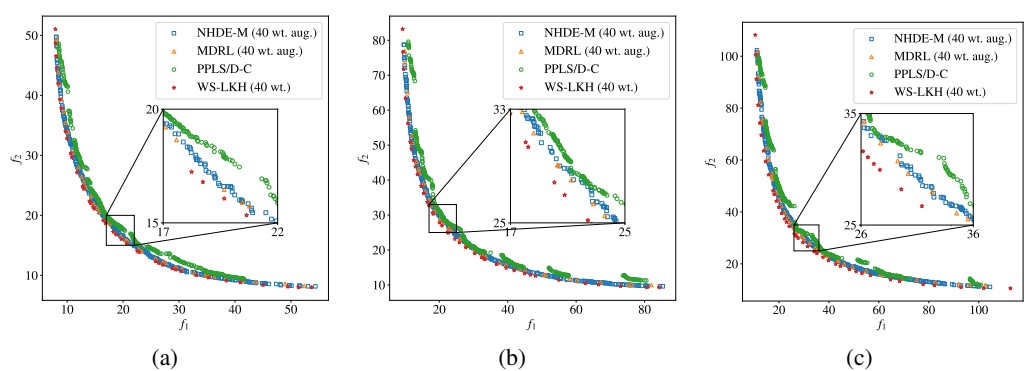

(a)        (b)        (c)

Figure 6: Pareto fronts of NHDE-M and compared methods on benchmark instances. (a) KroAB100. (b) KroAB150. (c) KroAB200.

in Figure 6. As can be clearly observed, NHDE-M exhibits superior generalization capability to the state-of-the-art MOEA and other neural methods with regard to the convergence and diversity.

## H Detailed results of NHDE-P on benchmark instances

Table 8 records the detailed results of NHDE-P and other baselines on benchmark instances, which demonstrate the superiority of NHDE-P.

## I Details of compared methods in ablation study

In Figure 4(a), we compare NHDE-P with decomposition-based DRL without indicator (NHDE w/o I) and indicator-based DRL without decomposition (NHDE w/o D) to study the effect of the indicator-enhanced DRL. Concretely, NHDE w/o I removes the HV indicator in the reward and the Pareto front graph in the inputs. NHDE w/o D dispenses with weights, removes the scalar objective in the reward, and only adopts the HV indicator to guide the model. In each subproblem without a weight, a new solution (or multiple sampled solutions) is produced to maximize the HV indicator

Table 8: Results of NHDE-P on benchmark instances.

| Method | HV↑ | KroAB100 |NDS|↑ | Gap↓ | Time | HV↑ | KroAB150 |NDS|↑ | Gap↓ | Time | HV↑ | KroAB200 |NDS|↑ | Gap↓ | Time |
|---|---|---|---|---|---|---|---|---|---|---|---|---|---|
| WS-LKH (40 wt.) | **0.7007** | 40 | **-0.47%** | 53s | **0.6989** | 39 | **-0.92%** | 1.9m | **0.7404** | 40 | **-1.58%** | 2.2m |
| PPLS/D-C (200 iter.) | 0.6785 | 388 | 2.71% | 31m | 0.6659 | 441 | 3.84% | 1.1h | 0.7100 | 491 | 2.59% | 3.1h |
| DRL-MOA (101 models) | 0.6903 | 67 | 1.02% | 10s | 0.6794 | 72 | 1.89% | 18s | 0.7185 | 73 | 1.43% | 23s |
| PMOCO (40 wt.) | 0.6862 | 36 | 1.61% | 7s | 0.6802 | 37 | 1.78% | 9s | 0.7174 | 38 | 1.58% | 12s |
| NHDE-P (40 wt.) | 0.6926 | 179 | 0.69% | 8s | 0.6873 | 225 | 0.75% | 12s | 0.7247 | 246 | 0.58% | 16s |
| PMOCO (40 wt. aug.) | 0.6916 | 38 | 0.83% | 8s | 0.6861 | 37 | 0.92% | 11s | 0.7223 | 39 | 0.91% | 16s |
| NHDE-P (40 wt. aug.) | 0.6974 | 317 | 0.00% | 11s | 0.6925 | 365 | 0.00% | 13s | 0.7289 | 377 | 0.00% | 17s |

Table 9: Results with change of the number of weights.

| Number of wt. | | Time | | HV | | |NDS| | |
|---|---|---|---|---|---|---|---|
| NHDE-P | PMOCO | NHDE-P | PMOCO | NHDE-P | PMOCO | NHDE-P | PMOCO |
| 10 | 150 | 14s | 14s | 0.6334 | 0.6355 | 76 | 56 |
| 20 | 300 | 26s | 26s | 0.6373 | 0.6359 | 103 | 63 |
| 40 | 600 | 53s | 53s | 0.6388 | 0.6361 | 127 | 68 |
| 80 | 1200 | 1.3m | 1.4m | 0.6395 | 0.6362 | 146 | 70 |

Table 10: Results for the runtime proportion of each module.

| Module of NHDE-P | Runtime Proportion |
|---|---|
| PMOCO | 13% |
| Indicator-enhanced inference | 5% |
| MPO | 82% |

under the current Pareto front. The surrogate landscape cannot be defined due to the disuse of weights, so the current whole Pareto front is taken as the input to the model.

Since HV can comprehensively measure convergence and diversity, indicator-based NHDE w/o D should find a Pareto set with good overall performance in intuition. However, it is even inferior to decomposition-based NHDE w/o I in practice. This fact reveals that it is difficult for the deep model to learn to construct solutions to directly optimize HV due to the high complexity of HV.

With respect to NHDE w/o MPO, which is used to evaluate the impact of our MPO strategy, it still samples multiple solutions for each subproblem, but only the solution with the maximum reward is preserved according to the view of single-objective optimization.

## J Runtime analysis

We provide the NHDE and PMOCO results on Bi-TSP50 with similar runtime, by changing the number of weights used in both methods in Table 9. When using a few weights (short runtime), PMOCO is slightly better than NHDE with close runtime (since the increasing number of weights can rapidly raise the performance of PMOCO). However, when using more weights ($N \geq 300$ for PMOCO), NHDE is consistently better than PMOCO with a close runtime. This further verifies that increasing weights may not effectively produce more Pareto solutions for existing neural solvers, while our NHDE can boost the limitation of such decomposition-based neural solvers, especially in diversity. The superiority is more significant for larger and more complex MOCVRP instances.

Moreover, we present the runtime proportion of each module in NHDE-P, including the original PMOCO, indicator-enhanced DRL, and MPO. The experiment is conducted on Bi-TSP50, as shown in Table 10. Please note that the runtime of the indicator-enhanced DRL during inference is mainly spent by the heterogeneous graph attention (HGA). The indicator-enhanced inference costs 5% runtime, with the complexity of the attention mechanism in PMOCO being $O(n^2)$ and the additional complexity in HGA being $O(nK)$. MPO costs 82% runtime, since the update of Pareto front needs more computation, i.e., $O((K + J)J)$.

Table 11: Comparison on NHDE-P with NHDE w/o I when using much more weights.

| Method | HV | |NDS| | Time |
|---|---|---|---|
| NHDE-P (600 wt.) | 0.6405 | 177 | 6.3m |
| NHDE w/o I (600 wt.) | 0.6372 | 141 | 5.9m |

We observe that the promising performance of NHDE comes more from MPO (see NHDE w/o MPO with HV 0.6335 in Figure 4(a)) than the indicator-enhanced inference (see NHDE w/o I with HV 0.6361 in Figure 4(a)). As presented in Table 10, the indicator-enhanced inference only costs a very small part of runtime, while MPO costs most of the runtime. Thus, their corresponding contributions to performance are reasonable, considering their compurational efforts.

We also compare NHDE-P with NHDE w/o I (equivalent to PMOCO with MPO) when using much more weights (i.e., 600) on Bi-TSP50. As shown in Table 11, our NHDE outperforms PMOCO with MPO, where PMOCO spends similar runtime to NHDE with the extra MPO module.

## K  Weight assignment

Recall that we use uniformly distributed weights during inference, which is a mainstream method for weight assignment when no information about the Pareto front is known in advance. For decomposition-based methods, weight assignment methods may affect the solution distribution. However, our NHDE can better alleviate this issue compared with pure decomposition-based neural heuristics for two reasons: (1) NHDE can generate more diverse solutions as verified by our experiments. (2) NHDE can also flexibly handle arbitrary weights during inference, enabling it to integrate seamlessly with proper weight assignment methods. When knowing the approximate scales of different objectives beforehand, we can first normalize them into [0,1] to derive a more uniform Pareto front. Otherwise, we can assign biased and non-uniform weights during inference to obtain more uniformly distributed solutions.

We present the results on Tri-TSP with asymmetric Pareto fronts, as shown in Figure 7. For Tri-TSP instances, the coordinates for the three objectives are randomly sampled from $[0, 1]^2$, $[0, 0.5]^2$, $[0, 0.1]^2$, respectively. The results show that non-uniform weights, which are obtained by multiplying uniform weights by (1,2,10) element-wise and then normalizing them back to $[0, 1]^3$, can produce a relatively more uniform Pareto front. Besides, compared with PMOCO (see Figure 7(b)), NHDE-P (see Figure 7(c)) can enhance diversity, thereby alleviating the non-uniform distribution of solutions.

## L  Hyperparameter study

We further study the effects of $N'$ (the number of weights used in training), $K$ (the limited size of the surrogate landscape of the Pareto front), and $J$ (the limited number of *points* from new solutions for updating the Pareto front).

We present the results of various $N'$ on Bi-TSP50 in the table below. As shown in Figure 8(a), $N' = 5$ and $N' = 10$ cause inferior performance, while proper $N'$ ($20 \leq N' \leq 40$) results in desirable performance. Intuitively, when limiting the same total gradient steps in training, a larger $N'$ means fewer instances used for model training. In this sense, too large $N'$, i.e., with insufficient instances, could lead to the inferior performance for solving unseen instances. On the other hand, too small $N'$ could prevent the model from learning favorable weight representations and thus deteriorate the final performance. Hence we choose $N' = 20$ in this paper.

Figure 8(b) displays the results of various values of $K$, where $K = 20$ is a desirable setting. When $K$ is too small, some key information of the Pareto front would be lost, thereby degrading the performance. When $K$ is too large, the deep model cannot cope with numerous points of the Pareto front, also leading to deterioration of the performance.

We provide the HV and runtime of NHDE-P on Tri-TSP50 with the changed $J$ in Figure 8(c). We observe that by limiting $J$, the massive time of the update can be curtailed with only a little sacrifice of performance. Since $J = 200$ is a good trade-off between HV and runtime, we use it in this paper.

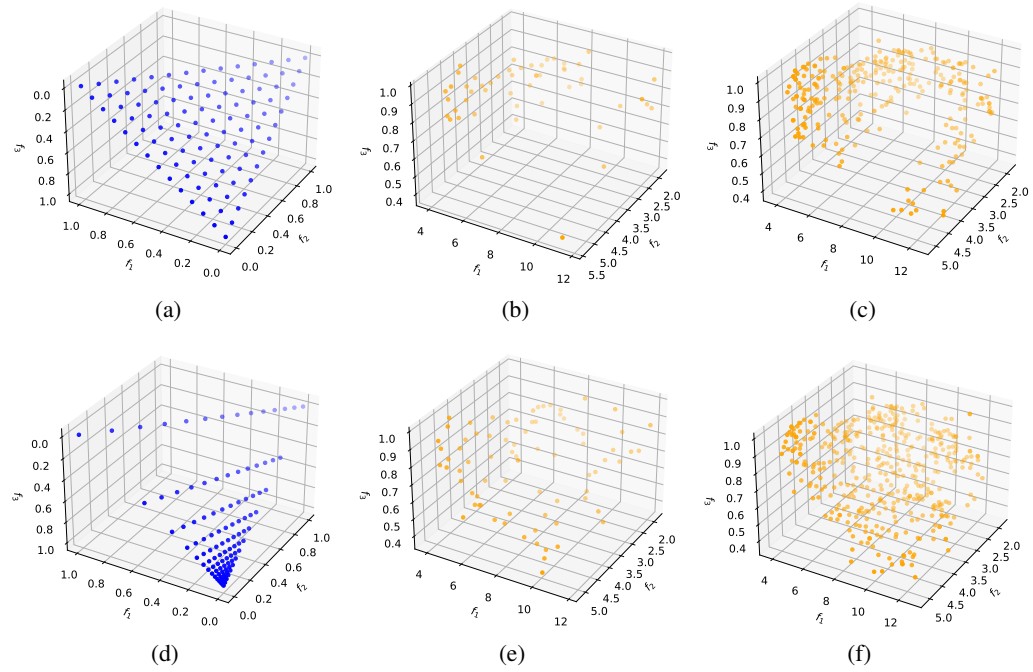

Figure 7: Solutions generated by using 105 uniform/non-uniform distributed weights on an instance of Tri-TSP20 with asymmetric Pareto front. (a) Uniform weights. (b) PMOCO with uniform weights. (c) NHDE-P with uniform weights. (d) Non-uniform weights. (e) PMOCO with non-uniform weights. (f) NHDE-P with non-uniform weights.

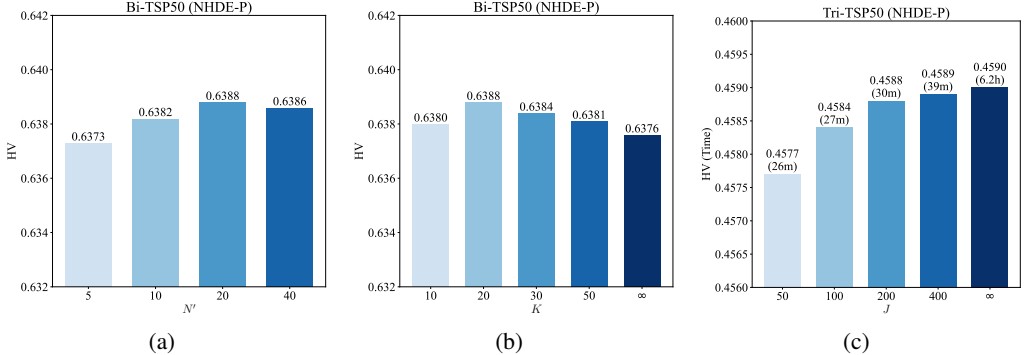

Figure 8: Hyperparameter study. (a) Effect of the number of weights used in training. (b) Effect of the limited size of the surrogate landscape of the Pareto front. (c) Effect of the limited number of *points* from new solutions for updating the Pareto front.

## M Additional analysis

In the inference, diversity factors are linearly changed, which means different emphasis between the scalar objective and the HV indicator. We test other settings of the diversity factors, e.g., some fixed values. As shown in Figure 9(a), different settings of the diversity factors have almost no impact on the performance, except $\boldsymbol{w}^1 = \cdots = \boldsymbol{w}^N = (1, 0)$ only emphasizing on HV. A possible reason is that the deep model is not good at learning the mapping from diversity factors to the complicated reward involving HV.

Recall that NHDE solves the subproblems dependently, and we simply use the shuffled weights. To study the effect of the random shuffle, we execute independent 10 runs. The boxplot of the results is

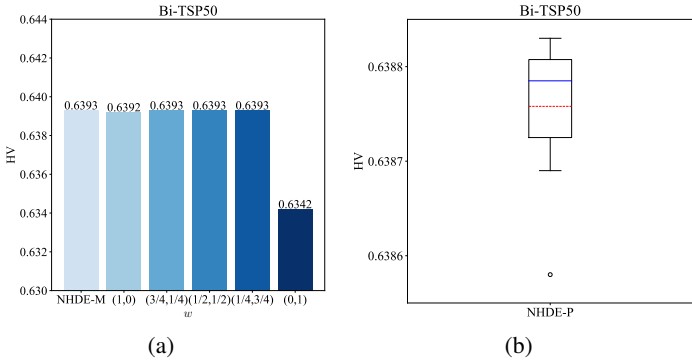

Figure 9: Additional analysis. (a) Effect of diversity factors. (b) Effect of shuffled weights.

Table 12: Results for the number of duplicated solutions.

| Method | \|DS\| |
|---|---|
| NHDE-P (40 wt.) | 196 |
| NHDE w/o I (40 wt.) | 225 |

Table 13: Variances of the methods.

| Method | Bi-TSP20 | | Bi-TSP50 | | Bi-TSP100 | |
|---|---|---|---|---|---|---|
| | HV | Variance | HV | Variance | HV | Variance |
| WS-LKH (40 wt.) | 0.6266 | $3.24 \times 10^{-4}$ | 0.6402 | $1.59 \times 10^{-4}$ | 0.7072 | $4.82 \times 10^{-5}$ |
| PPLS/D-C (200 iter.) | 0.6256 | $3.45 \times 10^{-4}$ | 0.6282 | $1.72 \times 10^{-4}$ | 0.6844 | $6.13 \times 10^{-5}$ |
| DRL-MOA (101 models) | 0.6257 | $3.31 \times 10^{-4}$ | 0.6360 | $1.67 \times 10^{-4}$ | 0.6970 | $5.09 \times 10^{-5}$ |
| PMOCO (40 wt.) | 0.6258 | $3.31 \times 10^{-4}$ | 0.6331 | $1.64 \times 10^{-4}$ | 0.6938 | $5.08 \times 10^{-5}$ |
| PMOCO (600 wt.) | 0.6267 | $3.28 \times 10^{-4}$ | 0.6361 | $1.55 \times 10^{-4}$ | 0.6978 | $4.71 \times 10^{-5}$ |
| NHDE-P (40 wt.) | 0.6286 | $3.19 \times 10^{-4}$ | 0.6388 | $1.58 \times 10^{-4}$ | 0.7005 | $4.76 \times 10^{-5}$ |
| PMOCO (40 wt. aug.) | 0.6266 | $3.26 \times 10^{-4}$ | 0.6377 | $1.60 \times 10^{-4}$ | 0.6993 | $4.99 \times 10^{-5}$ |
| PMOCO (100 wt. aug.) | 0.6270 | $3.28 \times 10^{-4}$ | 0.6395 | $1.54 \times 10^{-4}$ | 0.7016 | $4.81 \times 10^{-5}$ |
| NHDE-P (40 wt. aug.) | 0.6295 | $3.14 \times 10^{-4}$ | 0.6429 | $1.52 \times 10^{-4}$ | 0.7050 | $4.73 \times 10^{-5}$ |

presented in Figure 9(b). As shown, the random shuffle of weights only exhibits a slight impact on the performance. A specialized order of the weights may raise the performance, but it is beyond the scope of this paper, which would be explored in the future.

We implicitly show the reduced duplicated solutions by the metric |NDS|, i.e., the number of non-dominated solutions. Empirically, more non-dominated solutions mean fewer duplicated solutions and fewer dominated solutions. Thus, the larger |NDS| values of our method (especially in comparison to the other neural solvers) indicate that our method can produce a smaller number of duplicates to some extent. Furthermore, we add another evidence to verify that the indicator-enhanced DRL can hinder duplicated solutions. Specifically, we directly report the average number of duplicated solutions (|DS|) of our NHDE-P and NHDE-P without indicator (NHDE w/o I) on Bi-TSP50 in Table 12. As can be seen, our NHDE-P using the HV indicator can effectively guide the model to generate fewer duplicated solutions. Intuitively, our model is trained to construct new Pareto solutions different from existing ones in the Pareto front, which could achieve higher HV in the reward.

To further verify our results are statistically significant, we have conducted a Wilcoxon rank-sum test at a 1% significance level for the results in all groups, which means our results are statistically significant. We additionally report the variances of the results in Table 13. All methods have small variances of hypervolumes, where our NHDE-P achieves the stablest performance.