# OpenReview forum: "Neural Multi-Objective Combinatorial Optimization with Diversity Enhancement"
_NeurIPS.cc/2023/Conference — NeurIPS 2023 poster_

### Official Review · Reviewer_ETpH · 2023-06-23

**Soundness:** 3 good
**Presentation:** 3 good
**Contribution:** 2 fair
**Rating:** 5
**Confidence:** 3

**Summary:**

The paper presents a novel neural heuristic approach, called NHDE, for Multi-Objective Combinatorial Optimization (MOCO) problems. The proposed approach enhances the diversity of Pareto solutions by introducing a multiple Pareto optima (MPO) strategy and an indicator-enhanced Deep Reinforcement Learning (DRL) method. Experimental results on classic MOCO problems demonstrate that NHDE outperforms state-of-the-art neural baselines, achieving superior overall performance with a higher diversity of Pareto solutions.

**Strengths:**

1.	The paper proposes a novel algorithm to generate diverse solutions for combinatorial optimization (CO) problems.
2.	The proposed method is able to enhance existing methods to achieve improvements on diversity metrics including hypervolumes.


**Weaknesses:**

1.	The solving time of proposed solver is relatively slow compared to other RL-based CO solvers. Besides, one of the curious results is that the solving time of LKH is much slower than the time in previous papers. (see question 1)
2.	The authors do not discuss the scope of solvers of proposed diversity enhancement (see question 2).


**Questions:**

1.	As far as I know, LKH is a pretty fast heuristic solver for TSP problem. However, the reported running time of LKH in experiments of TSP is extremely slow and much slower than that reported in previous papers (e.g. [2]). Could authors explain this?
2.	From my understanding, the proposed method is able to conduct two tasks at a same time: (1) solve the CO problem (2) generate diverse solutions. The supported CO problems includes TSP, VRP. What is the scope of CO problems that the proposed diverse enhancement is able to handle? Does the diverse enhancement able to enhance all RL-based solvers? What about other neural solvers that do not use RL (e.g. [3])?
3.	How does other recent neural solvers perform on solution diversity? (e,g. [4,5])
4. in Line 118, when does the sub-problem required, for the CO solving itself, or diversity enhancement for multiple evaluation metrics? (Why does there N sub-problems instead of M?)

[1] Kwon Y D, Choo J, Kim B, et al. Pomo: Policy optimization with multiple optima for reinforcement learning[J]. Advances in Neural Information Processing Systems, 2020, 33: 21188-21198.

[2] Xin L, Song W, Cao Z, et al. NeuroLKH: Combining deep learning model with Lin-Kernighan-Helsgaun heuristic for solving the traveling salesman problem[J]. Advances in Neural Information Processing Systems, 2021, 34: 7472-7483.

[3] Wang R, Yan J, Yang X. Combinatorial learning of robust deep graph matching: an embedding based approach[J]. IEEE Transactions on Pattern Analysis and Machine Intelligence, 2020.

[4] Kwon Y D, Choo J, Kim B, et al. Pomo: Policy optimization with multiple optima for reinforcement learning[J]. Advances in Neural Information Processing Systems, 2020, 33: 21188-21198.

[5] Qiu R, Sun Z, Yang Y. DIMES: A Differentiable Meta Solver for Combinatorial Optimization Problems[C]//Advances in Neural Information Processing Systems.


**Limitations:**

It could be better if the authors could discuss potential limitations of the paper.

---

> ### Author Rebuttal · Authors · 2023-08-10
>
> Thanks for your valuable comments. Firstly, **we clarify the scope of this paper, since there seems to be a misunderstanding by the reviewer**. We focus on Multi-Objective CO (MOCO), a domain different from the Single-Objective CO (SOCO), even though the latter can employ diversity enhancement. MOCO seeks a set of Pareto-optimal solutions, each signifying a unique trade-off between various conflicting objectives rather than optimizing them separately as in SOCO  (see Figure 3 for illustration). Hence, solving MOCO is much harder than solving SOCO. We refer the reviewer to Section 3 for the definitions and details about MOCO.
>
> **To W1: More runtime than RL-based CO solvers.** We would like to note that although our NHDE-P takes longer runtime than the original RL-based PMOCO, in our original paper, we compared our NHDE-P not only with the original PMOCO but also its enhanced version for fairness (we adjusted the number of preferences in the enhanced PMOCO, extending its runtime, which, in some cases, even surpassed our runtime). In practice, it is worthy to find more significantly profitable solutions with acceptable prolonged time. As illustrated in Table 1, our method consistently outperforms, even under the modified conditions, e.g., ours (0.00%, 1.5h) outperforms PMOCO's (3.17%, 4.2h) on Tri-TSP100.
>
> We have further supplemented results on Bi-TSP50 with more choices on the number of preferences, as presented in the table below. When using a few preferences (short runtime), PMOCO is slightly better than NHDE with close runtime. However, when using more preferences ($N\geqslant 300$ for PMOCO), NHDE  is consistently better than PMOCO with close runtime. This further verifies that increasing runtime may not effectively produce more Pareto solutions for existing neural solvers, especially in diversity. Note that our superiority on larger size (i.e.,100) and more complex CVRP would be more significant.
>
> HV: NHDE-P/PMOCO|Time: NHDE-P/PMOCO|Pref.: NHDE-P/PMOCO
> :-:|:-:|:-:
> 0.6334/0.6355|14s/14s|10/150
> 0.6373/0.6359|26s/26s|20/300
> 0.6388/0.6361|53s/53s|40/600
> 0.6395/0.6362|1.3m/1.4m|80/1200
>
> **To W2/Q1: Runtime of WS-LKH.** We clarify that the reported time of WS-LKH (LKH with decomposition using 40 preferences) is reasonable because: 1) decomposition-based MOCO methods need to solve the single-objective CO subproblems multiple (i.e., 40) times, and 2) the practical solving time varies with the hardware.
>
> We give three specific examples to better illustrate the runtime. For 200 instances of Bi-TSP100, WS-LKH (40 pref.) with 10000 trials costs 2.7 hours in our experiments, so each single-objective subproblem for an instance spends 2.7$\*$3600/40/200=1.2 (s). **Example 1:** In NeuroLKH[2], LKH with 1000 trials costs 938 seconds for 1000 instances of single-objective TSP100, so each TSP100 instance with 10000 LKH trials spends about 938/1000$\*$10=9.4 (s). **Example 2:** In AM, LKH with 10000 trials using 32 virtual CPUs expends 21 minutes for 10000 instances of TSP100, so each TSP100 instance spends about 21$\*$60/10000$\*$32=4.0 (s). **Example 3:** In [7], WS-LKH for 200 instances of Bi-TSP100 consumes 3.1 hours.
> ```
> [6] Attention, Learn to Solve Routing Problems!, ICLR, 2019.
> [7] Pareto Set Learning for Neural Multi-Objective Combinatorial Optimization, ICLR, 2022.
> ```
>
> **To W3/Q2: Clarify the scope.** Our studied scope is MOCO, which aspires both desirable optimality and diversity in the Pareto front beyond SOCO with diversity enhancement. **[Scope of CO]** Our NHDE has the potential to solve most CO defined on graphs including routing, scheduling, bin packing, etc, since we adopt a GNN as policy model. For other COs, one can replace the deep model with corresponding ones while keeping our diversity enhancement schemes for MOCO including indicator-enhanced DRL and multiple Pareto optima strategy. **[Scope of solver]** We agree that NHDE is generic to boost existing decomposition based RL solvers for MOCO. We believe it can also be applied to other neural solvers that do not use RL by modifying the training algorithm. However, existing neural MOCO solvers almost use RL, since the label (the exact Pareto front) of MOCO is extremely hard to obtain (even more diffcult than NP-hard SOCO). Note that non-RL neural solve [3] referred by the reviewer is used to solve SOCO setting only.
>
> **To Q3: Solution diversity.** Firstly, unlike the study [4,5] that focused on finding a single optimal solution for optimizing a single objective by exploring diverse solutions, we emphasize that the studied MOCO pursues a set of Pareto solutions captured by both optimality and diversity for optimizing multiple conflicting objectives. We note that the diversity schemes used in these SOCO methods (e.g., POMO and DIMFES) may not be directy used for optimizing multiple conflicting objectives without MOCO techniques such as the decomposition. Nevertheless, we agree that they can inspire future works to further enhance the diversity of the single-objective sub-problem decomposed in MOCO, and we will cite and discuss this in the revised paper.
>
> **To Q4: Why sub-problem required?** Since MOCO optimizes multiple objectives, it pursues a set of Pareto-optimal solutions representing trade-off among the conflicting objectives, rather than outputting solutions for each $M$ objectives separately. Thus, the decomposition-based methods adopt $N$ preferences (usually $N>>M$) to scalarize each MOCO problem into multiple SOCO problems, which correspond to $N$ subproblems with various trade-off. The set of Pareto solutions can be derived by solving the $N$ subproblems, which can approximately solve the original MOCO.
>
> **To Limitation.** We have discussed the limitation in Conclusion, i.e., the HV calculation during training needs extra computational cost, thereby impeding the application of NHDE on many-objective problems. One potential solution is to consider efficient HV approximations to boost the training efficiency in the future.

---

> > ### Comment · Reviewer_ETpH · 2023-08-15
> >
> > To authors,
> >
> > My main concerns have been addressed, and I would raise my score to 5.

---

> > > ### Author Response · Authors · 2023-08-15
> > >
> > > We thank the reviewer very much for reviewing our response and recognizing our work.

---

### Official Review · Reviewer_DbfB · 2023-06-27

**Soundness:** 2 fair
**Presentation:** 3 good
**Contribution:** 2 fair
**Rating:** 5
**Confidence:** 4

**Summary:**

This paper proposes a number of techniques to improve the diversity of solutions in neural multi-objective combinatorial optimization (MOCO): 1) solving preference-conditioned subproblems dependently by taking a subset of the current Pareto front as input to a graph attention-based policy network; 2) using the hypervolume indicator as the reward; and 3) sampling multiple solutions from the policy for each subproblem to update the Pareto front. Previous decomposition-based MOCO methods are augmented with these techniques, and empirically evaluated on multi-objective TSP, CVRP, and KP benchmark problems. Results show that there is improvement in average hypervolume compared to previous work, and ablations show that each design choice has positive contribution to average performance.

**Strengths:**

Originality: This paper's ideas of using a reward based on the hypervolume and using the current Pareto front as part of the state input to the policy are novel (to my knowledge). Doing so ensures a stationary environment for RL, in contrast to previous methods that use distance-based rewards without observing the global solution space that leads to nonstationarity for RL.

Quality: The paper covers most of the important details in regards to problem motivation, background and related work, description of methods and analysis of results.

Clarity: The technical writing and notation is clear and concise.

Significance: The topic of multi-objective CO is an important one that arises in practice and there is increasing work on learning-based approaches, so this is a relevant and significant paper for the conference.

**Weaknesses:**

The abstract claims that the proposed techniques are intended to "hinder duplicated solutions for different subproblems." Even though the results show an increase in average diversity, as measured by the hypervolume, the paper does not provide concrete and direct evidence of this specific claim of reducing number of duplicates.

The method relies on at least three hyperparameters---$N$, the number of subproblems and preferences used in training; $K$, the number of top solutions from the current Pareto front to use as input to the policy; and $J$, the number of new solutions to be sampled for updating the Pareto front. The main paper does not provide any sensitivity study or guidance on how to choose them in practice, so it is unclear whether the proposed techniques are robust or brittle.

The results in Table 1, Table 2, and Figure 4 only report the average performance, but do not show the variance of results. It is not possible to determine whether the results are statistically significant.

The proposed techniques rely on a base method, which has some specific limitations:
1. The overall approach relies on the base method of sequentially solving subproblems, each determined by a preference. It is not clear how the ordering of subproblems impacts the performance of the method.
2. The policy is a graph attention network, so it requires the CO problem instances to be defined on a graph, so it's not clear the method generalizes to other CO problems where there is no natural graph structure.

In Section 4.1, where the paper defines the state space, it is not clear why the preferences are not part of the state, if the policy is expected to act differently for different subproblems with different preferences.

Other minor issues to be fixed or clarified:
- Lines 32-33 and the claim on lines 35-36 need citations.
- There are some typographical and grammatical issues: e.g.
  - line 19-20: "..., practical yet more complex"
  - line 233 is an incomplete sentence
  - line 257 "detials"
- In line 119, the subscripts $1,\dotsc,T$ appear without explanation. Readers have to guess that it denotes the sequence representation of the solution.
- From line 147, one can guess that each solution is represented by its objective values as an $M$-dimensional vector, but this should be explicitly stated.
- In the results, the best $|\text{NDS}|$ should also be in bold.

**Questions:**

The authors should:
- show that results are statistically significant
- provide evidence for the claim that the techniques reduce the number of duplicate solutions to subproblems
- explain how to choose the three hyperparameters and show sensitivity results

**Limitations:**

The use of a graph attention model limits the method to problems with a graph.

---

> ### Author Rebuttal · Authors · 2023-08-10
>
> Thanks for the valuable comments.
>
> **W1: Reduced duplicated solutions.**  We implicitly show the reduced duplicated solutions by the metric &#124;NDS&#124;, i.e., the number of non-dominated solutions. Empirically, more non-dominated solutions mean fewer duplicated solutions and fewer dominated solutions. Thus, the larger NDS values of our method (especially in comparison to the other neural solvers) indicate that our method can produce a smaller number of duplicates to some extent. We will further clarify the above point in our paper.
>
> Furthermore, we have added another evidence to verify that the indicator-enhanced DRL can hinder duplicated solutions for different subproblems. Specifically, we directly report the average number of duplicated solutions (|DS|) of our NHDE-P and NHDE-P without indicator (NHDE w/o I) on Bi-TSP50 in the table below. As can be seen, our NHDE-P using the hypervolume indicator can effectively guide the model to generate fewer duplicated solutions.
> Intuitively, our model is trained to construct new Pareto solutions different from existing ones in the Pareto front, which could achieve higher HV  in the reward. To make it sufficient, we will add the experiment with analyse in our paper.
>
> |Method|&#124;DS&#124;|
> |---|:---:|
> |NHDE-P (40 pref.)|196|
> |NHDE w/o I (40 pref.)|225|
>
> **W2:  Sensitivity study.** First of all, we conducted sensitivity analysis of $K$ in our submission (the first paragraph of Appendix J). Despite that, we followed the suggestion and have added the analyses of $N$ (the number of preferences used in training, which corresponds to the symbol $N'$), and $J$ (the number of new solutions to be sampled for updating the Pareto front).
>
> We present the results of various $N'$ on Bi-TSP50 in the table below. As shown, $N'=5$ and $N'=10$ cause inferior performance, while proper $N'$ ($20\leqslant N'\leqslant 40$) results in desirable performance. Intuitively, when limiting the same total gradient steps in training, a larger $N'$ means fewer instances used for model training. In this sense, too large $N'$, i.e., with insufficient instances, could lead to the inferior performance for solving unseen instances. On the other hand, too small $N'$ could prevent the model from learning favorable preference representations and thus deteriorate the final performance. Hence we choose $N'=20$ in this paper.
>
> |$N'$|HV|
> |---|:---:|
> |5|0.6373|
> |10|0.6382|
> |20|0.6388|
> |40|0.6386|
>
> We provide the HV and runtime of NHDE-P (210 pref. aug.) on Tri-TSP50 with the changed $J$. We observe that by limiting $J$, the massive time of the update can be curtailed with only a little sacrifice of performance. Since $J=200$ is a good trade-off between HV and runtime, we use it in this paper.
>
>
> |$J$|HV|Time|
> |---|:---:|:---:|
> |50|0.4577|26m|
> |100|0.4584|27m|
> |200|0.4588|30m|
> |400|0.4589|39m|
> |6400 (unlimited)|0.4590|6.2h|
>
> For $K$, we refer to Appendix J for the experiment with analyses. In summary, we choose $K=20$ for a better trade-off between the state representation of Pareto front and computational effort of the deep model.
>
> **W3:Variance of results.**  Following the suggestion, we test the variances of the results on Bi-TSP20 and report them in the table below. All methods have small variances of hypervolumes, where our NHDE-P achieves the stablest performance. Due to the space limitation here, we will present the variances on all problems in the final version.
>
> |Method|HV|Variance|
> |---|:---:|:---:|
> |WS-LKH (40 pref.)|0.6266|$3.24\times 10^{-4}$|
> |PPLS/D-C (200 iter.)|0.6256|$3.45\times 10^{-4}$|
> |DRL-MOA (101 models)|0.6257|$3.31\times 10^{-4}$|
> |PMOCO (40 pref.)|0.6258|$3.31\times 10^{-4}$|
> |PMOCO (600 pref.)|0.6267|$3.28\times 10^{-4}$|
> |NHDE-P (40 pref.)|0.6286|$3.19\times 10^{-4}$|
> |PMOCO (40 pref. aug.)|0.6266|$3.26\times 10^{-4}$|
> |PMOCO (100 pref. aug.)|0.6270|$3.28\times 10^{-4}$|
> |NHDE-P (40 pref. aug.)|0.6295|$3.14\times 10^{-4}$|
>
> To further verify our results are statistically significant, we also conducted a Wilcoxon rank-sum test for the results in Table 1, Table 2, and Figure 4 in the main paper. We observe that the best and second-best results achieved by our method are both statistically different from the results by the other methods, at a 1% significance level in all groups, which means our results are statistically significant.
>
> **W4: Sequence of subproblems.** In fact, we also noticed the order of subproblems could impact the performance, and studied this effect in Appendix J (see the 3rd paragraph) in our submission. We independently executed 10 runs with the random shuffle of preferences, and the result indicates the order only exhibits a slight impact on the performance. Specifically, most of HV values are 0.6388, and the worst (only one) HV value is 0.6386. We refer the reviewer to Figure 7(c) in Appendix J for the complete result. Please note that a highly specialized order of the preferences may raise the performance, but it is beyond the scope of this paper.
>
> **W5: Generalization.** As mentioned, our current model based on a graph neural network can only tackle CO problems defined on graphs. However, most CO problems (with their variants) can be actually defined on a graph, such as routing, scheduling, and bin packing. Moreover, it is natural and straightforward to adapt our model to solve other CO problems with different data structures, via replacing the graph neural network with the specialized neural network for any specific CO problem. We hope our work could inspire such future work on various types of CO problems.
>
> **W6: Missing preference.** Thanks for pointing this out.  Actually, we input the preference (as a part of the state) into our deep model, as we described in Section 4.2. But we miss it when describing the state definition. We will add it in Section 4.1 for the coherence .
>
> **W7: Minor issue.**  We will polish the paper accordign to the suggestion.

---

> > ### Comment · Reviewer_DbfB · 2023-08-15
> > **Acknowledgement of rebuttal**
> >
> > I appreciate the author's detailed rebuttal and additional empirical results. The authors have made progress in addressing my main concerns regarding hyperparameter sensitivity and missing variance (i.e. statistical significance of results), and I hope to see all variances reported in the final version. I agree that other work can build on the insights here to implement models for CO problems with different structures. Regarding having preference as part of the state, it's not immediately clear from Section 4.2 due to the dense notations, so please do state it in the text. Increasing the score accordingly.

---

> > > ### Author Response · Authors · 2023-08-16
> > >
> > > We thank the reviewer very much for reviewing our response and recognizing our work. We will add the preference as a part of the state in Section 4.2 in the final version. Specifically, we will revise the description in lines 126-128 as "... the *state* includes the preference $\lambda^i$, the partial solution ...". Moreover, we will continually proofread and improve the paper according to your suggestion.

---

### Official Review · Reviewer_TPc1 · 2023-07-07

**Soundness:** 2 fair
**Presentation:** 3 good
**Contribution:** 3 good
**Rating:** 6
**Confidence:** 4

**Summary:**

This work proposes NHDE, a novel neural heuristic with diversity enhancement method, to solve multi-objective combinatorial optimization (MOCO) problems. The main contribution of NHDE is to overcome the crucial duplicated solution issue for the widely-used decomposition-based method, and generate a set of diverse Pareto solutions in a reasonable runtime. The proposed method has two major components: 1) an indicator-enhanced model structure with heterogeneous graph attention (HGA) to generate diverse Pareto solutions, and 2) a simple yet powerful multiple Pareto optima (MPO) strategy to identify more Pareto solutions for each decomposed subproblems.

The proposed NHDE method is flexible and can be incorporated into different neural MOCO approaches. Experimental results show that NHDE can achieve promising performances on various MOCO problems, such as multi-objective traveling salesman, capacitated vehicle routing, and knapsack problems.

**Strengths:**

+ This work is generally well-written and easy to follow.

+ Multi-objective combinatorial optimization (MOCO) is important for many real-world applications, and the decomposition-based method is a widely-used approach for MOCO. It is promising to see the proposed NHDE can efficiently tackle the crucial duplicated solution issue for the decomposition-based neural method.

+ NHDE can be flexibly incorporated into different decomposition-based methods, and it obtains promising performance on various MOCO problems.

**Weaknesses:**

I have the following concerns for the proposed method:

**1. Solution Distribution**

NHDE is a promising approach to tackle the duplicated solution issue for the decomposition-based method. Another important issue for the decomposition method is the shape of generated Pareto front, where the solution distribution will be very sensitive to the scalarization method, especially for problems with more than two objectives. In real-world applications, it is often hard to choose the proper scalarization method for a new problem before optimization. One concrete example can be found in Appendix D.5 in the PMOCO[1] for neural MOCO. Can NHDE also (partially) tackle this issue? It is interesting to know the shape of generated Pareto front by NHDE on the tri-TSP with different scalarization methods.

[1] Pareto Set Learning for Neural Multi-Objective Combinatorial Optimization, ICLR 2022.

**2. Batch Inference and Subproblem Order**

It seems that NHDE needs to sequentially generate the solutions for different subproblems one by one (even for a single instance) since it takes the current Pareto front as input, while PMOCO and MDRL can generate solutions in batch. Is it possible to design a batch version for NHDE? Will the order of subproblems significantly affect NHDE's performance, especially for problems with more than two objectives?

**3. Runtime Analysis**

NHDE requires longer runtime than PMOCO and MDRL for the same number of preferences. It is interesting to know NHDE's performance with a limited runtime budget (e.g., a few seconds) similar to PMOCO. What is the runtime cost of the indicator-enhanced inference and MPO in NHDE separately?

**4. Effect of MPO**

The ablation study in Figure 4 is very helpful, and I believe it is for Bi-TSP50. According to the results, it seems that the performance of NHDE w/o MPO (0.6335) is similar to the PMOCO (0.6331) but requires a longer runtime. Does the promising performance of NHDE mainly come from MPO but not the indicator-enhanced inference? What is the performance of PMOCO(600 pref.) with MPO?

**Questions:**

- Please address the concerns raised in Weaknesses.

- It is better to also report the results of NHDE + MDRL (NHDE-M) in the main paper.

**Limitations:**

The limitation of the proposed method is discussed in the conclusion section.

---

> ### Author Rebuttal · Authors · 2023-08-10
>
> Thanks for the valuable comments.
>
> **W1: Solution Distribution.** We agree with the reviewer that the scalarization may affect the solution distribution. Our NHDE can better alleviate this compared to pure decomposition-based methods for two reasons:
> * NHDE can generate more diverse solutions as verified by our experiments.
> * NHDE can flexibly deal with arbitrary preferences during inference, thus is able to adopt proper scalarization methods.
>     * When knowing the approximate scales of different objectives beforehand, we can first normalize them into [0,1] to derive a more uniform Pareto front.
>     * NHDE can adopt preference assignment method used in PMOCO (see Appendix D.5 in the original old version of PMOCO paper) to obtain a more uniform Pareto front.
>
> We have supplemented the results on tri-TSP with asymmetric Pareto fronts, in the PDF under global response. For tri-TSP instances, the coordinates for the three objectives are randomly sampled from $[0,1]^2$, $[0,0.5]^2$, $[0,0.1]^2$, respectively. The results show that non-uniform preferences, which are obtained by multiplying uniform preferences by (1,2,10) element-wise and then normalizing them back to $[0,1]^3$, can produce a relatively more uniform Pareto front. Besides, compared with PMOCO, NHDE-P can enhance diversity, thereby alleviating this issue.
>
> **W2: Batch Inference and Subproblem Order.**  NHDE could not accomplish batch inference for subproblems, since NHDE dependently solves subproblems sequentially. However, NHDE itself has achieved batch inference for instances, which can reduce the time for solving a set of instances. Furthermore, we note that the run time of NHDE is acceptable since it can generate a desirable Pareto front with diverse solutions by using relatively fewer preferences, which means a lighter sequential time.
>
> Moreover, though PMOCO and MDRL can achieve batch inference for subproblems essentially, it is still hard to realize it. For a batch of subproblems, PMOCO needs to generate a batch of parameters in the neural network, while MDRL needs to be fine-tuned to derive a batch of neural networks for subproblems. Both of them could consume prohibited large GPU memory in practice.
>
> The order of subproblems would impact the performance, and we have studied this in Appendix J. We conducted 10 runs with randomly shuffled preferences and found minimal performance variation. Most HV results hovered around 0.6388, with the lowest at 0.6386. Figure 7(c) in Appendix J visualizes these results. While a tailored preference order might enhance performance, it is beyond the scope of this paper.
>
> **W3: Runtime Analysis.** We have added the NHDE and PMOCO results on Bi-TSP50 with similar runtime, by changing the number of preferences used in both methods. We present the comparison in the table below.  When using a few preferences (short runtime), PMOCO is slightly better than NHDE with a close runtime (since the increasing number of preferences can rapidly raise the performance of PMOCO). However, when using more preferences ($N\geqslant 300$ for PMOCO), NHDE  is consistently better than PMOCO with the close runtime. This further verifies that increasing preferences may not effectively produce more Pareto solutions for existing neural solvers, while our NHDE can boost the limitation of such decomposition-based neural solvers, especially in diversity. Our superiority on larger size (i.e.,100) and more complex CVRP would be more significant.
>
> HV: NHDE-P/PMOCO|&#124;NDS&#124;: NHDE-P/PMOCO|Time: NHDE-P/PMOCO|Pref.: NHDE-P/PMOCO
> :---:|:---:|:---:|:---:
> 0.6334/0.6355|76/56|14s/14s|10/150
> 0.6373/0.6359|103/63|26s/26s|20/300
> 0.6388/0.6361|127/68|53s/53s|40/600
> 0.6395/0.6362|146/70|1.3m/1.4m|80/1200
>
> Also, we have added the runtime proportion of each module in NHDE-P, including the original PMOCO, indicator-enhanced DRL, and MPO. The experiment is conducted on Bi-TSP50 (results shown in the table below). Please note that the runtime of the indicator-enhanced DRL during inference is mainly spent by the heterogeneous graph attention (HGA). The indicator-enhanced inference costs 5% runtime, with the complexity of the attention mechanism in PMOCO being $O(n^2)$ and the additional complexity in HGA being $O(nK)$. MPO costs 82% runtime, since the update of Pareto front needs more computaiton, i.e., $O((K + J)J)$.
>
> Module of NHDE-P|Runtime Proportion
> ---|:---:
> PMOCO|13%
> Indicator-enhanced inference|5%
> MPO|82%
>
> **W4: Effect of MPO.** We notice the case mentioned by the reviewer, where the effect of the indicator-enhanced inference on PMOCO is slight by comparing NHDE w/o MPO with PMOCO. However, to more accurately evaluate the effect of the indicator-enhanced inference on our NHDE (with HV 0.6388), NHDE without indicator enhancement (NHDE w/o I with HV 0.6361) should be involved in the comparison, as shown in Figure 4(a). In fact, NHDE w/o I in our implementation is equivalent to PMOCO equipped with MPO, which means that the indicator-enhanced inference leads to more improvement, i.e., 0.6388 over 0.6361, in this case.
>
> Despite this, we observe that the promising performance of NHDE comes more from MPO (see NHDE w/o MPO with HV0.6335 in Figure 4(a)) than the indicator-enhanced inference. As discussed in the previous response (**W3**), the indicator-enhanced inference only costs a very small part of runtime, while MPO costs most of the runtime. Thus, their corresponding contributions to performance are reasonable, considering their compurational efforts.
>
> Following the suggestion, we have added the comparative results of PMOCO with MPO (600 pref.) and NHDE-P (600 pref.) on Bi-TSP50. As shown in the table below, our NHDE outperforms PMOCO with MPO, where PMOCO spends similar runtime to NHDE with the extra MPO module.
>
> Method|HV|&#124;NDS&#124;|Time
> ---|:---:|:---:|:---:
> PMOCO w MPO (600 pref.)|0.6372|141|5.9m
> NHDE-P (600 pref.)|0.6405|177|6.3m
>
> **Q1:** We will move the results of NHDE + MDRL (NHDE-M) to the revised main paper.

---

> > ### Comment · Reviewer_TPc1 · 2023-08-15
> > **Thank you for the detailed response.**
> >
> > Thank you for your detailed response, and all my concerns have been properly addressed. I keep my positive score (6) and lean toward accepting this paper.

---

> > > ### Author Response · Authors · 2023-08-15
> > >
> > > We thank the reviewer very much for reviewing our response and recognizing our work.

---

### Official Review · Reviewer_fppw · 2023-07-08

**Soundness:** 3 good
**Presentation:** 3 good
**Contribution:** 3 good
**Rating:** 6
**Confidence:** 3

**Summary:**

This paper focus on solving multi-objective combinatorial optimization by neural methods. The authors observe that decomposition-based neural MOCO methods may acquire similar solutions for neighboring subproblems. Therefore, to enhance the diversity of the non-dominated solution set, they introduce the hypervolume indicator to reward design and propose a heterogeneous graph attention mechanism that learns the interaction between the Pareto Front and the nodes of the instance. Besides, a multiple optimal strategy is proposed to sample and preserve more solutions. Experimental results show that the proposed NHDE generates more diverse solutions and achieves better hypervolume.

**Strengths:**

1. Hypervolume Indicator is firstly utilized to learn neural MOCO methods and this idea would inspire researchers to develop novel MOCO solvers beyond existing pure decomposition-based methods.
2. The diversity of the non-dominated solution set is a key point of MOCO problems. It is quite meaningful to enhance diversity.
3. Sufficient experiments are conducted, and the proposed method achieves better convergence and diversity.

**Weaknesses:**

The main concern is about the design of HGA. See the questions below.

**Questions:**

In lines 164-167, the authors claim that HGA can guide the constructed solutions distinct from existing ones. However, the proposed HGA learns the interaction between the node features (which represent the instance) and the points of Pareto Front (which are objective values), which could not represent the structure of any solutions. Could you explain why HGA can guide the diverse solution construction without explicitly considering the structure of the existing and current solutions?

**Limitations:**

Yes

---

> ### Author Rebuttal · Authors · 2023-08-10
>
> Thanks for the valuable comments.
>
> **To Question 1:** The reasons that HGA can guide the diverse solution construction without explicitly considering the structure of the existing and current solutions are as follows.
>
> 1. Intuition of implicitly considering the solution structure. Actually, the proposed neural network can use the attention computations to capture the favorable representation learning of the interaction between the node features (which represent the instance) and the points of Pareto front (which are objective values). Intuitively, the interaction contains the structure of solutions to some extent, considering the fact that Pareto front is the mapping from Pareto solutions, conditioned on the instance infomation. Therefore, the learned node and point embeddings implicitly contains the useful information of the existing and current solutions, by progressively  message passing and aggregating between the embeddings, so as to achieve the desirable final performance in our experiments. More specifically, the node-to-node attention reflects each node’s attention towards others for the construction of the current solution (i.e. instance conditioned impact on solution construction); the node-to-point attention reflects each node’s attention to points in the Pareto front for guidance of constructing solutions different from the existing solutions (i.e. instance-Pareto-front joint  impact on solution construction).
> 2. Reward driven diverse solution construction. We would like to note that another key design for produing differnt solutions is the reward design based on HV. As we know, this is first time HV is explicitly icluded in rewards for DRL method of multi-objective combinatorial optimization. Specifically, our reward design renders the current Pareto solution produced differently from existing ones in the Pareto front, for achieving higher HV in the final performance.
> 3. Complexity to explicitly embed all existing solutions. In our current design, the input of the neural network can be represented as a graph with variable numbers of points with a fixed low dimensions of the point features. Hence, it allows us to adopt a graph neural network, e.g., the graph attention network, to embed the point graphs. However, if we explicitly input the solution structure, which would be a graph with variable numbers of points with a variable high dimensions of the point features. Such embedding requires a quite complex neural network. Thus, we use the current design for a trade-off bewteen the performance and complexity.

---

> > ### Comment · Reviewer_fppw · 2023-08-16
> >
> > Thanks for your response. My concerns have been addressed. I will keep my score.

---

> > > ### Author Response · Authors · 2023-08-16
> > >
> > > We thank the reviewer very much for reviewing our response and recognizing our work.

---

### Author Rebuttal · Authors · 2023-08-10

Many thanks for all reviewers' constructive and valuable comments. Following their suggestions, we have made the following main revisions:

1. **Experiments:** We have provided more experimental results and analyses, including the solution distribution, runtime analysis, the evidence results about the reduced duplicated solutions, and more sensitivity analyses on hyperparameters.

2. **Significance:** We have added more analyses on the effect of MPO according to Reviewer TPc1 and the statistical results according to Reviewer DbfB.

3. **Additional questions:** We have explained some additional questions about the subproblem order, batch inference, runtime of the LKH solver, etc.

---

### Decision · Program_Chairs · 2023-09-21

**Decision:**

Accept (poster)

**Comment:**

This paper proposes an algorithm for generating diverse solutions for combinatorial optimization problems. The response addresses concerns raised by the reviewer, including questions about the scope of the proposed method and its runtime compared to other solvers, as well as the limitations of the approach. Overall, this paper presents a novel contribution to neural multi-objective combinatorial optimization.